# The H3K36me3 methyltransferase SETD2 contributes to PAF1C interactions with RNA Pol II and is required for neuronal differentiation

Christina Ambrosi[1,2], Ramon Pfaendler [1,2,4], Kristeli Eleftheriou [3], Stefan Butz[1,2], Davide Recchia [1,2,3], Xue Bao [3], Richard Cardoso da Silva [3], Niklas Kupfer [3], Ilse M Lagerwaard [3], Hanneke Vlaming [3], Nina Schmolka[1,5], Vivek Bhardwaj [3] & Tuncay Baubec [1,3]✉

## Abstract

**Chromatin modifications are essential for mammalian development, and their aberrant deposition is associated with human disease. While the mechanisms that deposit and remove histone modifications have been largely elucidated, their roles in regulating gene activity during cellular differentiation are yet to be fully understood. Here, we performed a deletion screen to identify stage-specific requirements of chromatin regulators during neuronal differentiation of mouse embryonic stem cells. We show that the H3K36me3 methyltransferase SETD2 is required for the establishment of neuronal gene expression during late stages of differentiation but is dispensable in mature neurons. Notably, this function is independent of its histone methyltransferase activity. Instead, SETD2 promotes interactions between the PAF1 complex and elongating RNA Pol II, suggesting a role in supporting efficient transcription of neuronal genes.**

**Keywords** Chromatin; H3K36me3; SETD2
**Subject Categories** Chromatin, Transcription & Genomics; Development

## Introduction

Chromatin modifications play important roles in orchestrating site-specific processes in a genomic context. Mutations in the enzymes that deposit these modifications and aberrant distribution of chromatin marks result in impaired development and are associated with numerous human diseases (Jones et al, 2016; Robertson and Wolffe, 2000). Thus, chemical marks on DNA and histones contribute to the establishment and maintenance of correct gene expression patterns and are essential for cellular identity and function. In the last decades, the genomic distribution of most chromatin marks has been characterized in various cell types and organisms, allowing us to better understand their potential involvement in regulating transcriptional programs (Mikkelsen et al, 2007; Lister et al, 2009).

Within the large panel of chemical modifications of chromatin, methylation marks on the histone H3 tails at lysine residues K9 or K27, and/or methylation of cytosines on the DNA are frequently associated with transcriptional inactivity of gene promoters and repetitive elements (Allis and Jenuwein, 2016; Villaseñor and Baubec, 2021). Besides these repressive modifications, tri-methylation of histone H3 at residue K36 (H3K36me3) is deposited to transcribed gene bodies. This bookmarking of transcribed genes is facilitated by the methyltransferase SETD2 through interactions with the elongating RNA Polymerase II (McDaniel and Strahl, 2017), resulting in the addition of a methyl group to histones modified by the mono- and di-methyltransferase activities of NSD1-3 and ASH1L (Shipman et al, 2024). The role of this bookmarking is not fully understood in mammals and several regulatory functions have been reported, including prevention of spurious transcription initiation from gene bodies (Carrozza et al, 2005), DNA damage response (Kanu et al, 2015), splicing regulation (Kolasinska-Zwierz et al, 2009), and chromatin cross-talk (Brien et al, 2012; Baubec et al, 2015). Loss of H3K36me3, either through mutations in *Setd2* or histone H3 K36M mutations, as observed in clear cell renal cell carcinoma (Dalgliesh et al, 2010; Ho et al, 2016) and chondrosarcoma (Behjati et al, 2013; Fang et al, 2016), respectively, hint to a potential role of this modification in preventing transformation and undifferentiated growth—highlighting its relevance for cellular function.

Interestingly, despite the functional requirement for SETD2 to catalyze the deposition of H3K36me3, ablation of *Setd2* in embryonic stem cells results in limited changes in gene expression and cellular identity, which is in stark contrast to the phenotypes observed in *Setd2*-KO mouse models (Baubec et al, 2015; Zhang et al, 2014; Hu et al, 2010). Similar outcomes were reported for

[1]Department of Molecular Mechanisms of Disease, University of Zurich, Zurich, Switzerland. [2]Life Science Zurich Graduate School, University of Zurich and ETH Zurich, Zurich, Switzerland. [3]Genome Biology and Epigenetics, Institute of Biodynamics and Biocomplexity, Department of Biology, Utrecht University, Utrecht, The Netherlands. [4]Present address: Institute of Molecular Systems Biology, ETH Zurich, Zurich, Switzerland. [5]Present address: Institute of Experimental Immunology, University of Zurich, Zurich, Switzerland. ✉E-mail: t.baubec@uu.nl

other important chromatin-modifying enzymes, including H3K27me3 or DNA methylation writers, where stem cells could survive in the absence of these marks, while their differentiation was impaired (Jackson et al, 2004; Boyer et al, 2006; Sakaue et al, 2010). This apparent cellular context-dependent role of chromatin modifications remains to be fully elucidated and requires systematic comparisons to test the contribution of individual chromatin modifications towards the establishment and maintenance of cell type-specific gene expression. Furthermore, recent studies highlight a role for histone-modifying enzymes that expands beyond their catalytic activities, where lack of catalytic activity was dispensable, questioning the essentiality of some of the histone modifications (Morgan and Shilatifard, 2023).

In this study, we compare the neuronal differentiation potential of isogenic mESC lines lacking the major chromatin regulators EED, DNMT1, DNMT3A, DNMT3B, and SETD2 to test their requirements for exit from pluripotency, lineage commitment, and terminal differentiation. We identify SETD2 as important for the establishment of neuronal gene expression patterns, but dispensable once these are established. We show that this function does not require the catalytic activity of SETD2, suggesting a non-catalytic role for this protein. By employing ChromID (Villaseñor et al, 2020) to identify the proteins associated with the elongating RNA Pol II in the absence and presence of SETD2, we reveal a reduced association of the PAF1 Complex with the elongating polymerase.

## Results

### Stage-specific requirement for chromatin regulators indicates a role for SETD2 in neuronal differentiation

To compare the contribution of DNA methylation, H3K27me3, and H3K36me3 to gene expression during cellular differentiation, we compiled a set of CRISPR-generated knock-out mouse embryonic stem cell (mESC) lines, including *Dnmt1,3a,3b* triple-KO ("TKO") (Domcke et al, 2015), *Eed*-KO (Villaseñor et al, 2020), and *Setd2*-KO (Baubec et al, 2015), all created in the same isogenic background (Appendix Fig. S1A). In addition, to test the combinatorial contribution of DNA methylation and H3K36me3, we ablated *Setd2* in the TKO background (quadruple-KO, "QKO") (Appendix Fig. S1B–E). All cell lines retained pluripotency and were able to proliferate in serum + LIF conditions without apparent changes in morphology, proliferation capacity, and lineage marker expression (Appendix Fig. S2A–E). Using these cell lines, we wanted to compare how the absence of these chromatin modifications influences exit from pluripotency and lineage commitment. Towards this, we differentiated the mESCs to neural progenitors and subsequently to maturing glutamatergic neurons in vitro (Bibel et al, 2004, 200) (Fig. 1A). As expected from previous studies (Zhang et al, 2014, 201; Jackson et al, 2004; Boyer et al, 2006; Sakaue et al, 2010) differentiation in cell lines lacking DNA methylation and H3K27me3 was severely compromised since *Eed*-KO cells did not survive exit from pluripotency (CAd4) and the *Dnmt*-TKO cells aborted differentiation at the neural progenitor stage (NPC) (Fig. 1A; Appendix Fig. S3A,B).

In contrast, we observed normal progression to the progenitor stage in the *Setd2*-KO mESCs, with similar numbers of surviving

cells and lack of morphological differences compared to wild-type cells (Fig. 1A; Appendix Fig. S3A). However, upon further differentiation to mature neurons, we observed that the survival of *Setd2*-KO-derived neural progenitors was strongly impaired, with around 20% developing into viable post-mitotic neurons expressing the neuronal marker Tuj1 (Fig. 1A,B; Appendix Fig. S3C). This impaired neuronal differentiation was reproduced in two independent knock-out clones and one constitutive *Setd2* knock-down cell line (Appendix Fig. S4A).

Following these observations, we wanted to explore if the reduced generation of neurons in the absence of SETD2 is due to impaired establishment or failure to maintain neuronal cell identity. We used a doxycycline-inducible sh*Setd2* mESC line to knock down *Setd2* at different timepoints during the neuronal differentiation (Appendix Fig. S4B). Cells that were treated constantly with doxycycline (day d0) or from early neural progenitor stages on (day d6 or d8) showed a low number of surviving and differentiated neurons, similar to the phenotype observed in the knock-out cells (Fig. 1C; Appendix Fig. S4C). In contrast, induction immediately after the early steps of neuronal differentiation (day d10) did not severely affect cell survival, resulting in 86-91% surviving neurons (Fig. 1C; Appendix Fig. S4C). Reduced levels of SETD2 in post-mitotic neurons after doxycycline-induced expression of shRNA at timepoint d10 were confirmed by RT-qPCR (Appendix Fig. S5A).

To corroborate these findings, we used an independent cell line where *Setd2* is endogenously tagged with the dTag degradation FKBP tag (Nabet et al, 2018). Addition of dTAG13 for 7 days results in a reduction of H3K36me3 (Appendix Fig. S5B), as shown previously for this cell line (Molenaar et al, 2022). We show that neuronal differentiation of *Setd2*-FKBP cell lines in the presence of dTAG13 results in 20% neuronal survival, while the addition of dTAG at day 10 of neuronal differentiation results in 90% survival (Appendix Fig. S5C,D). In addition, we performed washout experiments, where we start the neural differentiation from ESCs cultivated in the presence of dTAG13, and wash out the small molecule at day 0, 4, 6, and 7 of neuronal progenitor differentiation (Appendix Fig. S5E). While we observe that removal of the drug until day 4 is well-tolerated, removal at later timepoints of the differentiation results in reduced neuronal survival (Appendix Fig. S5E–G), indicating that SETD2 could be required during the late steps of neurodifferentiation. This is also visible from the flow-cytometric analysis of two neuronal surface proteins, CD24a (CD24) and CD56 (also known as NCAM1) (Schmolka et al, 2023), where we observe a reduction of CD24-positive neural progenitor cells in absence of SETD2 (50.6% in *Setd2*-KO vs. 65.1% in WT), and furthermore, an increase of the double negative (−CD24a/−CD56) cells that did not commit to neuronal lineage differentiation (24.2% vs. 11.4%, respectively) (Appendix Fig. S5H).

### SETD2 is required for the correct establishment of neuronal gene expression programs

To obtain further insight into the contribution of SETD2 or H3K36me3 to the establishment of neuronal gene expression programs, we performed bulk RNA-seq in wild-type and *Setd2-KO* neural progenitor cells (NPC) at day 8 of neuronal differentiation. Differential gene expression analysis at this early stage of neuronal commitment showed that 241 genes were upregulated, and 325

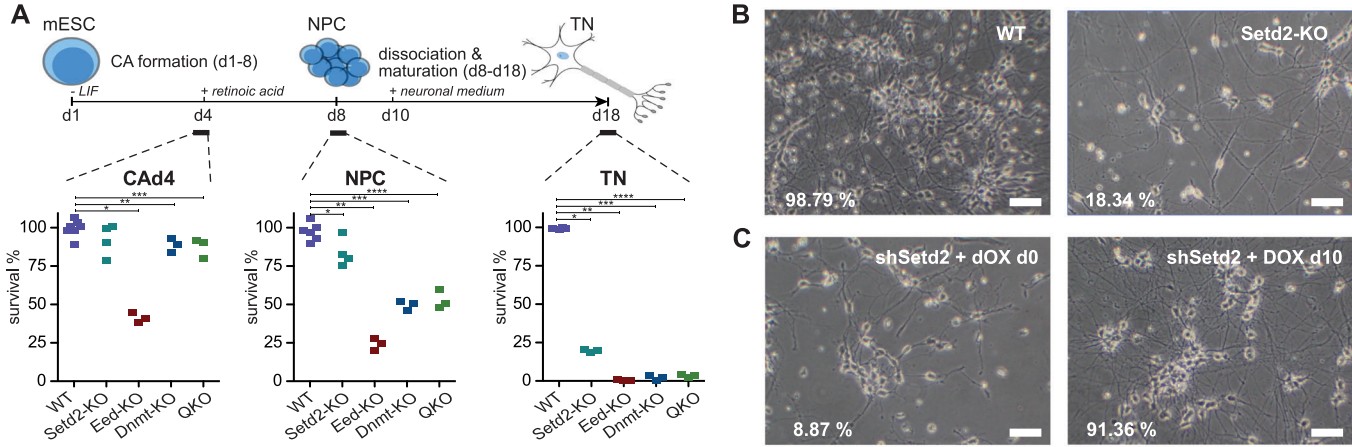

**Figure 1. Stage-specific requirement for chromatin regulators during exit from pluripotency, lineage commitment, and terminal differentiation.**

(A) Top: schematics depicting the neuronal differentiation protocol. Bottom: cell count assay using live-dead stain at the cellular aggregate stage day 4 (CAd4), neural progenitor cells (NPC), and terminal neurons stage day 14 (TN). Depicted are percentages of survival in WT, *Setd2*-KO, *Eed*-KO, *Dmnt*-TKO, and QKO cells, calibrated to 100% survival in WT cells. Data obtained from three to five independent replicates is shown. Asterisks indicate *P* values calculated using a two-tailed *t* test. For CAd4: *=5E-07, **=0.013, and ***=0.011; for NPC: *=0.012, **=1E-06, ***=2E-05, and ****=4E-05; for TN: *=2E-08, **=1E-09, ***=2E-08, and ****=1E-08. (B) Microscopy images of WT and *Setd2*-KO TNs at day d14 at ×100 magnification. Average percentages of survival after dissociation obtained from three independent experiments are indicated. (C) Same as in (B) but using TNs from Tet-inducible *Setd2* knockdown cell lines that were treated with 1 μg/ml doxycycline (DOX) from d0 to d14, or from d10 to d14. Continuous knockdown of *Setd2* results in neuronal cell death, while knockdown in mature neurons does not influence survival. Scale bars, 50 μm. Source data are available online for this figure.

genes were downregulated in the *Setd2*-KO neural progenitor cells (Fig. 2A; Dataset EV1). Gene set enrichment (GSEA) and gene ontology (GO) analysis showed high enrichment for neurodevelopmental-related processes in the downregulated gene set, suggesting a deregulation of neuronal transcriptional programs already at this stage (Appendix Fig. S6A–D). This was also confirmed by Motif Activity Response Analysis (Balwierz et al, 2014) which identified motifs preferentially bound by REST and SOX2 in the promoters of the downregulated genes (Fig. 2B). Genes showing increased gene expression were mainly related to developmental processes, reproduction and transcription, translation, and metabolic processes, and coincided with E2F1 and MYC/MAX transcription factor binding sites in their promoters, indicating increased proliferation and reduced differentiation (Fig. 2B; Appendix Fig. S6A,B,D). Furthermore, we observe that long genes and genes with long transcripts were more likely to be downregulated in the absence of SETD2 (Appendix Fig. S6E,F), which is also in accordance with the notion that neuronal-specific genes are longer than non-neuronal genes (Gabel et al, 2015; Zylka et al, 2015).

To examine whether these transcriptional changes are also related to splicing differences between wild-type and *Setd2*-KO, we performed alternative splicing analysis using rMATS-turbo (Wang et al, 2024). We identified a total of 18108 alternative splicing events to occur in the dataset. However, only 57 and 175 showed a significant increase or decrease in the *Setd2*-KO NPCs, respectively (Appendix Fig. S7A). Among those, skipped exons and retained introns were the most abundant, although without significant differences between the analyzed cell lines (Appendix Fig. S7A–C). Furthermore, we failed to find any correlations between the detected significant differential splicing events and H3K36me3 levels (Appendix Fig. S7D,E). This suggests that the transcriptional changes observed upon loss of SETD2 are not related to alternative

splicing, but rather that the changes in transcription, which also include splicing factors, could indirectly result in splicing differences (Appendix Fig. S7F).

To obtain a higher resolution view on the gene expression changes during the neuronal differentiation in the absence of SETD2, we performed single-cell RNA-sequencing from dissociated and sorted wild-type and *Setd2*-KO ESCs and NPCs (Muraro et al, 2016). Unsupervised clustering and with UMAP projection of all cells allowed us to identify ten different clusters (Appendix Fig. S8A,B, "Methods"). These clusters were grouped based on proliferation and neuronal markers into five groups representing progressive stages of ESC to NPC differentiation trajectory (Fig. 2C). While there is no clear separation of *Setd2*-KO and wild-type cells in groups associated with proliferating and undifferentiated cells, we observe a difference in gene expression state between the genotypes of cells as they advance in the differentiation process (Fig. 2D). This is also visible from differential expression analysis performed on unsupervised local neighborhoods of transcriptionally similar cells, where cells in more differentiated clusters show stronger differences in gene expression between *Setd2* KO and wild-type (Appendix Fig. S8C and Dataset EV2). When we calculate differential gene expression in separate neuronal and NPC neighborhoods, we detected 88 genes with different expression between *Setd2* KO and wild-type (Appendix Fig. S8D, "Methods"). For example, we find Myelin Transcription Factor 1 (Myt1), a factor that regulates Myelin gene expression and the transition to terminal differentiation (Nielsen et al, 2004; Vasconcelos et al, 2016) and Uncx, a transcription factor involved in the postmitotic differentiation of neurons (Sammeta et al, 2010; Rabe et al, 2012), to be downregulated in *Setd2*-KO (Fig. 2E). In addition, we find other markers relevant for neuronal differentiation as differentially expressed between *Setd2*-KO and wild-type NPC (Appendix Fig. S8D,E).

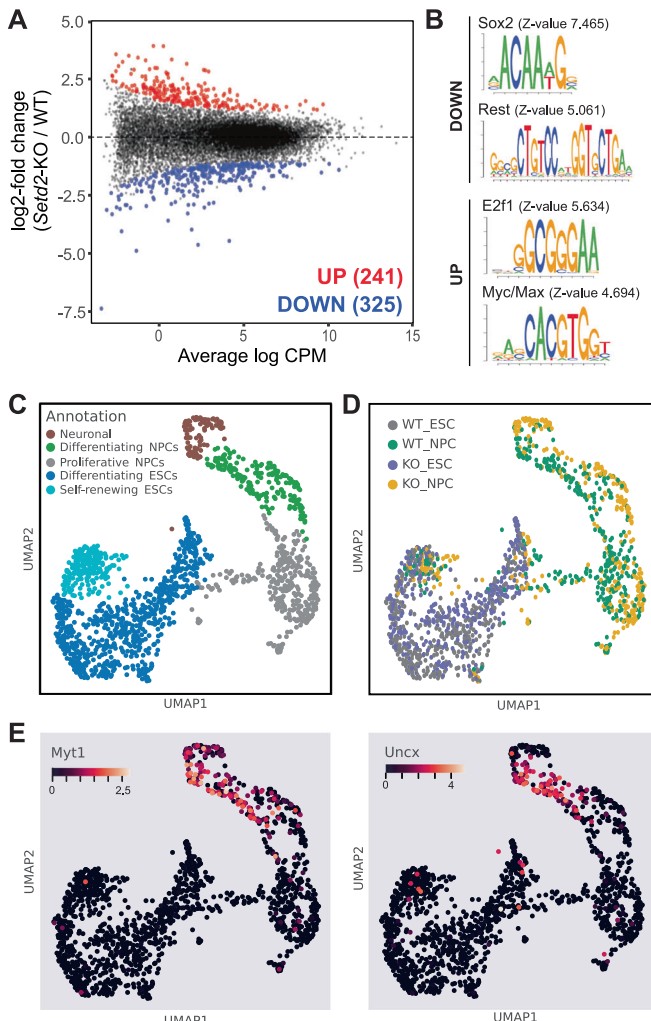

**Figure 2. SETD2 is required for the establishment of neuronal gene expression programs.**

(A) MA plot of bulk PolyA-RNA-seq results depicting differences in gene expression between wild-type (WT) and *Setd2*-KO NPCs (*P* value < 0.05, LFC > |1|). (B) Motif Activity Response Analysis identifies transcription factor motifs preferentially bound by Rest and Sox2 in the promoters of genes downregulated in *Setd2*-KO NPCs, and motifs coinciding with E2f1 and Myc/Max transcription factor binding in the promoters of upregulated genes. (C) UMAP projection of single-cell RNA seq data obtained from wild-type and *Setd2*-KO ESCs and NPCs. Shown are annotations of different cell states obtained from grouping unsupervised clusters shown in Appendix Fig. S8A. (D) Same UMAP projection as in (C) but showing the individual cell lines and cell types. (E) Expression of Myt1 and Uncx is predominantly observed in differentiating wild-type neuronal cells.

The results above indicate that the absence of SETD2 could influence the establishment of neuronal transcription, causing the observed differentiation failure. We argued that this deficit could be rescued by enforcing increased transcriptional initiation from neuronal genes. To test this, we generated *Setd2*-KO ES cell lines with *Tet*-inducible expression of the two bHLH transcription factors *Neurogenin 1* and *Neurogenin 2* (iNgn1/2) (Fig. 3A; Appendix Fig. S9A). Induced expression of these factors has been shown to enable rapid neurogenesis by driving genetic programs involved in the transition from stem cells to early neurons

(Busskamp et al, 2014). We induced their expression in *Setd2*-KO at the NPC stage and evaluated their ability to rescue incorrect establishment of neuronal gene expression programs by measuring the success of neuronal differentiation. Induction of Ngn1/2 indeed increased the survival rate of *Setd2*-KO neurons from 15.87 to 44.48%, compared to 88.25% in WT +dox (Fig. 3B; Appendix Fig. S9B). In addition, RT-qPCR analysis for selected lineage marker genes confirmed this partial rescue (Fig. 3C), suggesting that increasing the frequency of transcriptional initiation from neuronal genes through overexpressing neuronal master regulators can partially overrule the necessity for SETD2 in the differentiation process.

## Catalytic activity of SETD2 and H3K36me3 is largely dispensable for neuronal differentiation

While the above results indicate that SETD2 is required for the successful establishment of neuronal gene expression programs, they do not explain whether the observed phenotype is caused by the absence of the SETD2 protein, loss of SETD2 catalytic activity, or loss of H3K36me3. To address this, we first performed genome-wide chromatin measurements in wild-type and *Setd2*-KO neural progenitors. Besides the drastic removal of H3K36me3, we did not observe global changes in chromatin modifications or chromatin accessibility in the absence of SETD2 (Fig. 4A; Appendix Fig. S10A,B). Analysis of chromatin modifications at all gene bodies identified minor correlations with histone acetylation and anticorrelations with H3K4me3 in the absence of H3K36me3 (Fig. 4B; Appendix Fig. S10C). Increase in histone acetylation at gene bodies upon removal of H3K36me3 was previously reported in yeast (Carrozza et al, 2005; Joshi and Struhl, 2005; Keogh et al, 2005).

To evaluate if the changes in chromatin modifications coincide with the detected transcriptional changes in Setd2-KO NPCs, we focused our attention on the gene bodies and promoters of up and downregulated genes. While changes in RNA Pol II occupancy, chromatin accessibility, and acetylation correlated with differential transcription, we did not observe any relationship between differential gene expression and loss of H3K36me3 at gene bodies (Fig. 4C; Appendix Fig. S10D,E). For example, we observe that RNA Pol II occupancy is reduced at genes downregulated in *Setd2*-KO cells—in accordance with their reduced transcription (Fig. 4D). Concomitantly, similar trends were observed for chromatin accessibility and histone acetylation at the same set of genes in the absence of SETD2 (Appendix Fig. S10D,E). However, differential transcription occurred predominantly at genes with low H3K36me3 levels in their gene bodies (Appendix Fig. S10D,E). In addition, no significant changes were observed for H3K4me3 or H3K27me3 at the promoters of differentially expressed genes, suggesting that these marks are likely not involved in influencing transcription in the absence of SETD2 (Appendix Fig. S10D–F). We furthermore calculated the pausing index based on RNA Pol II occupancy at promoters and gene bodies for all genes, and for differentially expressed genes separately. We only observe a minor difference between wild-type and *Setd2*-KO cells, with genes downregulated showing a slightly reduced pausing index (Appendix Fig. S10G,H).

These results suggest an H3K36me3-independent role for SETD2 in establishing correct gene expression programs. To

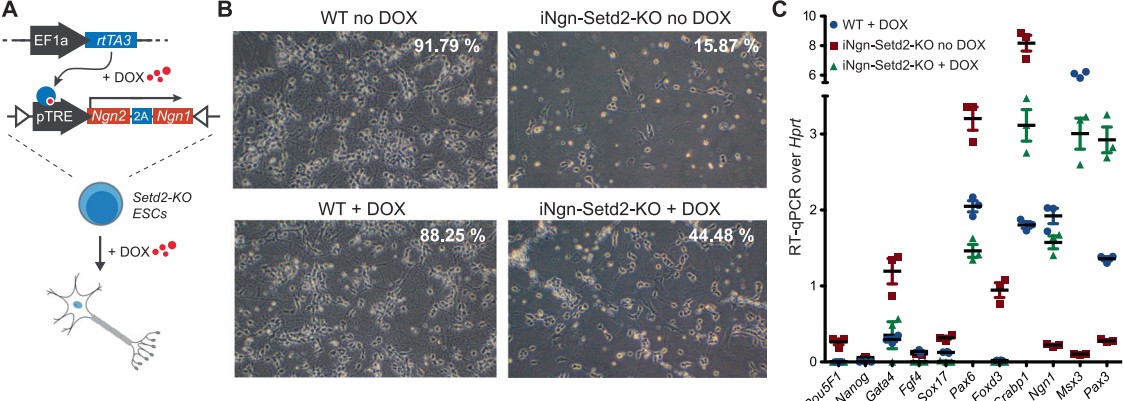

**Figure 3. Overexpression of master transcription factors overrides SETD2 requirement for neuronal differentiation.**

(A) Schematic overview of mESC line generation harboring a Tet-inducible Neurogenin 2-Neurogenin 1 (iNgn) master transcription factor cassette in a *Setd2*-KO background using recombination-mediated cassette exchange. (B) Microscopy images of in vitro-derived neurons (d14) show successful differentiation of neuronal cells derived from wild-type ESCs and partial rescue of cell death in *Setd2*-deficient cells by overexpression of an inducible Neurogenin 2-Neurogenin 1 fusion factor (iNgn). Similar results were obtained from two independent replicates. Cells were treated with 1 µg/ml doxycycline (+ DOX) during the entire in vitro differentiation, or with DMSO (no DOX). Mean neuronal survival rate (in percent) from three independent replicates is shown. (C) RT-qPCR of various lineage markers in WT and *Setd2*-KO NPCs, the latter expressing Neurogenin 1 and Neurogenin 2 upon 1 µg/ml doxycycline treatment (n = 3). Target genes: *Oct4* (*Pou5f1*), *Nanog* - embryonic stem cell markers; *Pax3, Pax6, Ngn1, Crabp1, Msx3, Foxd3* - neuronal marker; *Fgf4* - growth factor, *Sox17* - parietal endoderm marker. Gene expression was calibrated to *Hprt* - housekeeping gene expression. Bars and whiskers indicate means and SD. Source data are available online for this figure.

further test the requirement for the catalytic activity of SETD2 during neuronal differentiation, we engineered a point mutation in the catalytic SET domain (R1599H) (Sun et al, 2005) in both endogenous *Setd2* alleles (Appendix Fig. S10I). This point mutation resulted in a global reduction of H3K36me3 (Appendix Fig. S10J). Surprisingly, despite the global absence of H3K36me3, we observe successful differentiation of neuronal cells in presence of this catalytically inactive SETD2 enzyme (~65%, Fig. 4E). In addition, we cultivated wild-type ESCs in presence of the SETD2 inhibitor EPZ-719 (Lampe et al, 2021) at 1 µM concentration for two days, followed by neuronal differentiation under continued addition of the inhibitor and observed a similar outcome with minor to no effects on neuronal differentiation (Appendix Fig. S10K). Together, these results support the notion that catalytic activity of SETD2 and H3K36me3 is largely dispensable for neuronal differentiation.

## Reduced association of the PAF1 complex with the elongating RNA Pol II in the absence of SETD2

The results above suggest a catalytically independent role of SETD2 required during the establishment of neuronal gene expression. To test how SETD2 influences transcription, independent of H3K36me3, we investigated the protein interaction network of the elongating RNA Polymerase II in the absence of SETD2. Towards this, we fused the SRI domain of SETD2, which is responsible for binding the serine-2-phosphorylated CTD of RNA Pol II (Kizer et al, 2005), to the promiscuous biotin ligase TurboID (Branon et al, 2018). This enables the identification of proteins associated with the elongating RNA Pol II by proximity biotinylation. We first validated the correct colocalization of the SRI-TurboID probe to the elongating RNA Pol II by immunofluorescence microscopy of serine-2-phosphorylated RNA Pol II and biotin. The obtained colocalization coefficient suggests a strong correlation of biotin signals and elongating RNA Pol II in wild-type

cells expressing the SRI-TurboID probe, while cell lines expressing an unspecific NLS-TurboID probe lacking the SRI domain failed to show any colocalization (Appendix Fig. S11A,B).

We next introduced this probe in wild-type and *Setd2*-KO neural progenitor cells by recombinase-mediated cassette-exchange (Villaseñor et al, 2020) and performed biotin proximity-ligation followed by stringent protein enrichment as previously established (Villaseñor et al, 2020) (Fig. 5A; Appendix Fig. S11C). As a background control, we used the NLS-TurboID fusion protein lacking the SRI domain, expressed from the same genomic integration site (Appendix Fig. S11C). The enriched biotinylated proteins in wild-type and *Setd2*-KO neural progenitors were detected by quantitative, label-free liquid chromatography tandem mass spectrometry (LC–MS/MS), resulting in an average of 890 proteins collectively detected in all samples (Appendix Fig. S11D). Since the background NLS-TurboID samples did not show significant differences between the two genetic backgrounds (Appendix Fig. S11E), we pooled all replicates from the NLS-TurboID conditions and identified statistically-significant enriched proteins from the SRI-TurboID samples against this pooled background set. This resulted in 62 proteins significantly enriched in WT and 61 proteins enriched in *Setd2*-KO cells (Fig. 5B; Appendix Fig. S11F,G and Dataset EV3). In both samples, we identified proteins associated with GO Molecular Functions: RNA Pol II and RNA binding (Appendix Fig. S12A,Ba).

Among the RNA Pol II-interacting proteins, we identified several factors that were depleted in the absence of SETD2. These included proteins involved in transcription, splicing and chromatin regulation, according to their GO Molecular Function, such as KAT6A, RUXF, BRE1A or the catalytically inactive KMT2E (Fig. 5C–E; Appendix Fig. S12C). Interestingly, several factors of the core PAF1 complex (PAF1, CDC73, LEO1, CTR9) and related proteins involved in histone H2B Lys-120 monoubiquitination (RNF20, WAC), as well as elongation-associated proteins (CDK12, SUPT6, TCEB3, IWS1,

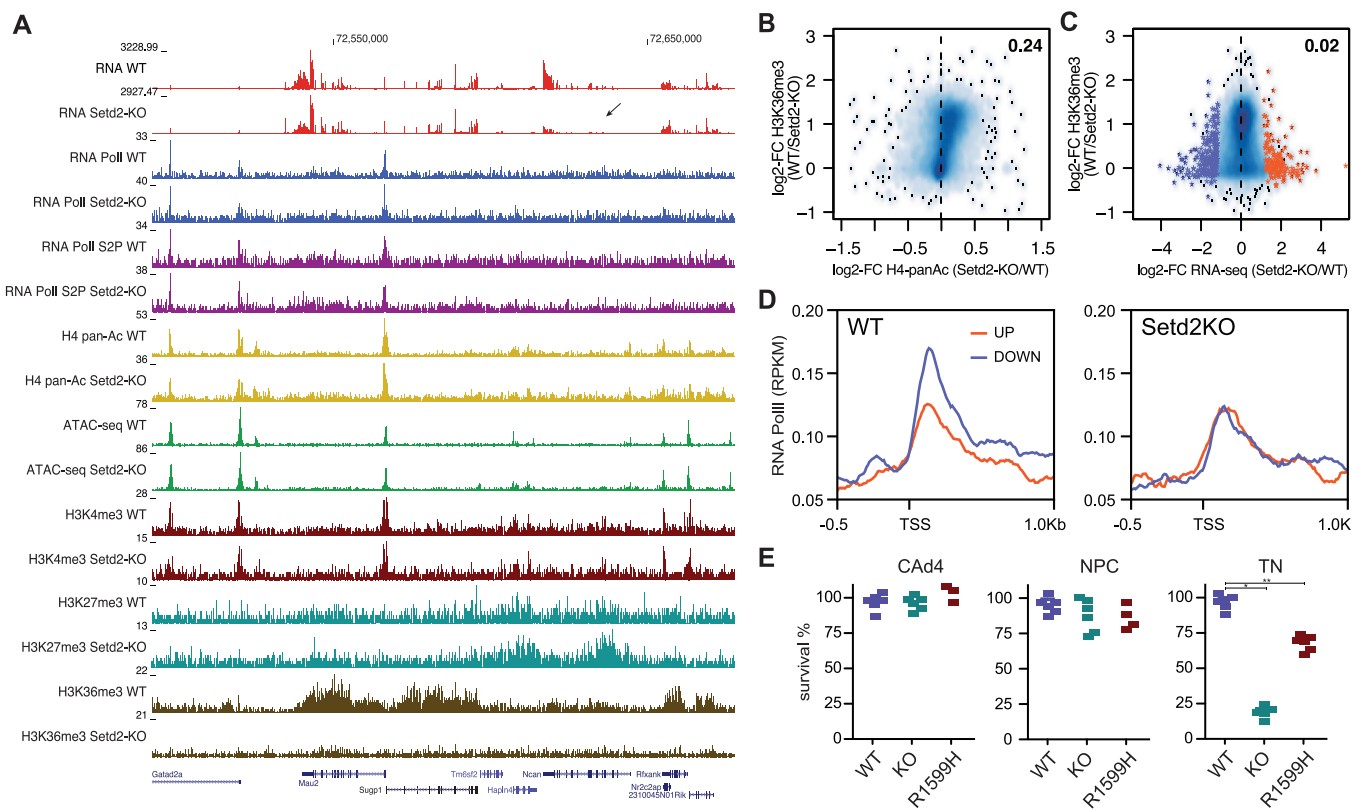

**Figure 4. Catalytic activity of SETD2 is dispensable for neuronal differentiation.**

(A) Representative genome browser view exemplifying differences in ChIP-seq signal of various chromatin marks and ATAC-seq signal between wild-type and *Setd2*-KO NPCs. Shown are read counts per 100 bp for ChIP-seq samples. Arrow exemplifies a downregulated neuronal gene *Ncan*. (B) Scatterplot showing a minor correlation between the measured H4-panacetylation changes in *Setd2*-KO over WT NPCs and H3K36me3 levels at gene bodies measured in WT cells. Pearson's correlation coefficient is shown. (C) Scatterplot showing changes in gene expression measured in *Setd2*-KO over WT NPCs and their relation to H3K36me3 levels at gene bodies measured in WT cells. Pearson's correlation coefficient is shown. (D) Metaprofile plots of RNA Pol II ChIP-seq data obtained from wild-type and *Setd2*-KO NPCs at genes up- and downregulated in the absence of SETD2. Downregulated genes show reduced Pol II signals in the absence of SETD2. (E) Cell survival assay using live-dead stain at CAd4, NPC, and TN stage (d14). Depicted are percentages of survival for WT, *Setd2*-KO, and SET domain catalytic mutant (R1599H) cells. Bars and whiskers indicate means and SD. Asterisks indicate *P* values calculated using a two-tailed *t* test: *=4E-011 and **=2E-06. Source data are available online for this figure.

PHF3) were depleted in absence of SETD2 (Fig. 5D). PAF1C plays a crucial role in transcriptional elongation and regulates the deposition of H2BK120ub (via RNF20/40) and H3K79me2 at transcribed genes (Van Oss et al, 2017). Indeed, we observe a reduction in global H3K79me2 levels, and a minor reduction in H2BK120ub in the absence of SETD2 (Appendix Fig. S13A), suggesting that the association of elongation factors with RNA Pol II could be directly or indirectly diminished in the absence of SETD2. To test if such changes in elongation factor interactions with the RNA Pol II would impact neuronal differentiation, we generated cell lines with dox-inducible shRNA expression against Paf1 and further Rnf20 (Appendix Fig. S13B). In both cases, we observe a strong effect on differentiation after reduction of RNF20 and PAF1 levels, with only 15–20% terminal neuronal differentiation potential, and resembling the results obtained in the absence of SETD2 (Appendix Fig. S13C,D).

## Discussion

By directly comparing the requirement for DNA methylation, H3K27me3, and H3K36me3 during exit from pluripotency and

neuronal differentiation, we show their requirement for re-wiring of transcriptional programs at different stages of differentiation. Most notably, and in contrast to DNA methylation and H3K27me3, SETD2 and H3K36me3 are dispensable for exit from pluripotency and neuronal lineage commitment. SETD2, however, is required during later stages of neuronal differentiation, where only 10–20% of the differentiated *Setd2*-KO neurons survive. This mirrors observations from KO experiments in mice, where *Setd2*−/− is embryonically lethal only at day E10.5, leading to growth defects including forebrain hypoplasia and unclosed neural tubes (Hu et al, 2010). By using inducible depletion of SETD2 at different stages of neuronal differentiation, we further show that SETD2 is required during the terminal differentiation stage of NPCs, while removal of SETD2 in differentiated neurons did not influence their survival. This suggests that SETD2 could play a role during the establishment of neuronal gene expression programs rather than maintenance in differentiated neurons. This was further corroborated by transcriptional defects in neuronal gene expression that were already observed in *Setd2*-KO NPCs from bulk and single-cell RNA-seq data, despite showing no apparent phenotypes. Interestingly, the single-cell RNA-seq data indicated that transcriptional

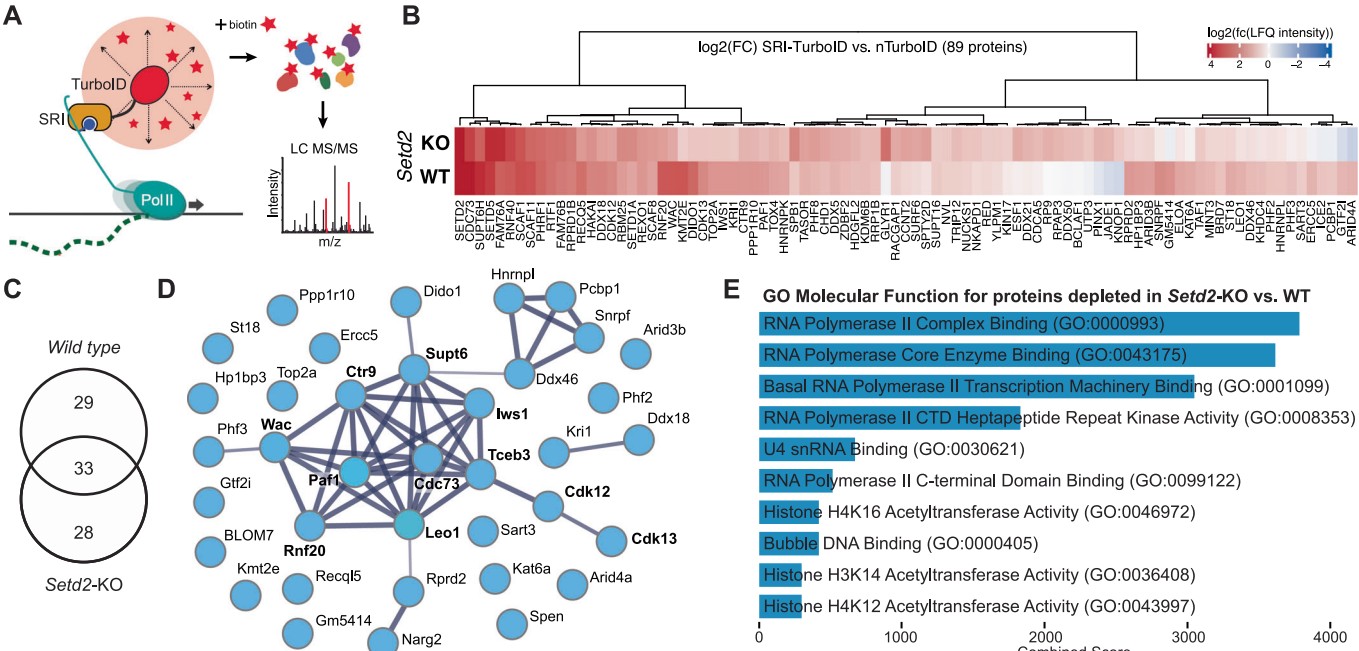

**Figure 5.   Loss of SETD2 results in reduced association of elongation factors with RNA Pol II.**

(A) Schematic overview of the ChromID setup using SETD2-SRI domain fused to TurboID to detect protein-protein interactions on the elongating RNA Pol II in wild-type and *Setd2*-KO NPCs using mass spectrometry. (B) Heatmap indicating the combined significantly enriched proteins in the wild-type and *Setd2*-KO NPCs sample over a background control expressing NLS-TurboID. Results obtained from four independent replicates. Colors indicate log2-fold change in LFQ intensity from SETD2-SRI over NLS-TurboID in the respective experiments. SETD2 peptide counts display combined tryptic peptides from endogenous SETD2 and SRI-TurboID. (C) Venn diagram indicating the overlap between proteins significantly enriched by SRI-TurboID over nTurbo in Setd2-KO and wild-type cells. (FDR-corrected two-tailed *t* test: FDR = 0.01, s0 = 0.1, $\log_2$ FC > 0, n = 4 independent replicates). (D) STRING analysis of proteins that show reduced RNA Pol II interactions in *Setd2*-KO NPCs when compared to wild-type NPCs identifies factors involved in RNA Pol II elongation. (E) Bar plot representing the top ten Molecular Function GO terms summarizing the proteins depleted from RNA Pol II in *Setd2*-KO NPCs. Data was generated using the Enrichr database. Source data are available online for this figure.

differences between wild-type and Setd2-KO NPCs were more evident during later stages of differentiation. There, we observed important markers related to the transition from progenitor to post-mitotic genes, such as Pax3, Pax6, Epha3, Myt1, or Uncx, to be differentially regulated in the absence of SETD2. To test if we can overcome the differentiation deficiency caused by the SETD2 depletion, we forced increased expression of neuronal genes through inducing ectopic expression of the master transcription factors Neurogenin 1 and Neurogenin 2 in *Setd2*-KO NPCs. This led to the re-expression of several neuronal marker genes and increased generation of fully differentiated neurons, indicating that the absence of SETD2 can be compensated.

H3K36me3 has been proposed to play important roles in regulating transcriptional elongation, gene body chromatin, or to prevent spurious transcription initiation from gene bodies (McDaniel and Strahl, 2017). By comparing genome-wide measurements for chromatin states and transcription in wild-type and *Setd2*-KO NPCs, we were not able to associate the observed changes in gene expression with the loss of H3K36me3. However, RNA Pol II occupancy at genes downregulated in Setd2-KO cells was reduced at promoters and gene bodies, in agreement with reduced gene expression. At the same time, genes that lost H3K36me3 in the absence of SETD2 did not display any changes in transcription or major differences in RNA Pol II occupancy. This lack of correlation points to a potential non-catalytic role of SETD2, which we further substantiated by showing that neurons expressing a catalytic dead

SETD2 or neurons grown in the presence of the SETD2 inhibitor EPZ-719 were able to differentiate. This finding is in line with a previous study, which showed that H3.3K36A mutations in both H3.3 genes are compatible with neuronal differentiation of ESCs and that the resulting changes in H3K36me3 do not necessarily alter gene expression (Gehre et al, 2020). Non-catalytic roles for histone-modifying enzymes have been previously described, indicating that cellular functions of chromatin modifiers can extend beyond their histone-modifying capacity (Morgan and Shilatifard, 2023). For example, the catalytic activity of the H3K4me1 methyltransferases MLL3-MLL4 has been reported to be dispensable for naive to primed transition of mESC, which is blocked in full KO cells (Dorighi et al, 2017).

To investigate the mechanism underlying the non-catalytic role of SETD2 during neuronal differentiation, we fused the SRI domain of SETD2 that binds to the Ser2-phosphorylated CTD of RNA Pol II (Kizer et al, 2005) to the promiscuous biotin ligase TurboID in order to investigate the protein interactome of the elongating RNA Pol II in the presence and absence of SETD2. Proximity biotinylation, followed by quantitative mass spectrometry, identified 90 significantly enriched proteins associated with RNA Pol II transcription, the elongation and splicing machinery, and chromatin regulation. Among these factors, we identified four out of five core subunits of the Polymerase Associated Factor 1 Complex (PAF1C; PAF1, CTR9, CDC73, and LEO1) and the PAF1C-associated factors SUPT6, IWS1, WAC, RNF20 to be depleted in

Setd2-KO NPCs. PAF1C interacts with the elongating RNA Pol II in gene bodies and is essential for co-transcriptional processes and Pol II processivity (Francette et al, 2021; Hou et al, 2019). Absence of PAF1C results in decreased H3K36me3 (Chu et al, 2007) and H2B ubiquitination (Ng et al, 2003), further connecting the activity of this complex with histone modifications at transcribed genes. Along these lines, we also observe a reduction of RNF20, the E3 ligase responsible for H2B ubiquitination in mammals, as well as WAC, a factor involved in linking RNF20/40 with the elongating RNA Pol II (Kim et al, 2005; Zhang and Yu, 2011), resulting in reduced H2BK120ub levels in Setd2-KO NPCs.

These results suggest that the absence of SETD2 could directly or indirectly influence the association of PAF1C with the elongating polymerase, potentially attenuating productive transcriptional elongation (Appendix Fig. 13E). How SETD2 contributes to the association of PAF1C with the elongation machinery is unclear. One speculation could be that it directly contributes as a structural component that helps to stabilize PAF1C by interacting with the S2-phosphorylated Pol II CTD via its C-terminal SRI domain, while it's N-terminal IDR, which has been identified to form condensates (Bhattacharya et al, 2021), could contribute to the stability of the elongation complex. Along these lines, PHF3, another S2-phosphorylated-CTD interacting protein, has been suggested to influence elongation through regulating phase separation of phosphorylated Pol II (Appel et al, 2021). Alternatively, SETD2 could influence PAF1C association with the elongating RNA Pol II indirectly. This could be via regulating interaction factors or RNA Pol II, such as SUPT6 and IWS1, that we have identified to be depleted from RNA Pol II in the absence of SETD2. SUPT6 has been shown to facilitate recruitment of PAF1C to RNA Pol II and is required for PAF1C's association with elongating RNA Pol II (Aoi et al, 2022; Vos et al, 2018), while the SUPT6 partner protein, IWS1, has been suggested to connect SETD2 and SUPT6 (Yoh et al, 2008). Alternatively, transcriptional changes in the absence of SETD2 could influence the presence and therefore the stoichiometry of such factors, resulting in reduced interactions with RNA Pol II. Along these lines, a recent study profiled nuclear proteomes from wild-type and Setd2-KO U2OS cells and identified a global reduction of RNA Pol II components and factors involved in elongation, including PAF1C members, in the absence of SETD2 (Kopczyńska et al, 2026). Future biochemical studies will allow to address the exact role of SETD2 in mediating PAF1C interactions with the elongating machinery.

Taken together, we suggest a novel, non-catalytic role of SETD2 in promoting transcription of neuronal genes during differentiation and promoting the association of the PAF1 complex with the elongating RNA Pol II. Although the PAF1C interaction with RNA Pol II is only mildly affected in the absence of SETD2, this could influence the productive transcription of long neuronal genes, linking the two observations together. Previous studies identified a role for SETD2 in promoting ESC to endoderm differentiation (Zhang et al, 2014), and Setd2 loss-of-function mutations or deletions are frequently found in renal cell carcinoma and associated with increased metastasis (The Cancer Genome Atlas Research Network, 2013; Duns et al, 2010). In addition, heterozygous mutations in Setd2 cause developmental delay, intellectual

disability, brain deformities, and macrocephaly (Luscan et al, 2014; Wu et al, 2023; Xu et al, 2021). Our study identifies novel roles of SETD2 in promoting neuronal gene expression, offering novel insights into potential mechanisms underlying these diseases.

# Methods

### Reagents and tools table

| Reagent/resource | Reference or source | Identifier or catalog number |
|---|---|---|
| **Experimental models** | | |
| HA36CB1 mouse ESC | Baubec et al, 2013 | |
| HA36CB1 Setd2-KO | Baubec et al, 2015 | |
| HA36CB1 DNMT-TKO | Domcke et al, 2015 | |
| HA36CB1 EED-KO | Manzo et al, 2017 | |
| HA36CB1 Setd2-DNMT-QKO | This study | |
| HA36CB1 Setd2-FKBP | Molenaar et al, 2022 | |
| **Recombinant DNA** | | |
| pX330-U6-Chimeric_BB-CBh-hSpCas9 | Addgene | 42230 |
| pRR-Puro | Addgene | 65853 |
| Ngn1/2_rtTA3 | Addgene | 61472 |
| **Antibodies** | | |
| Anti-Biotin | Sigma-Aldrich | 3737373 |
| Anti-RNAP2-S2P | Abcam | Ab5095 |
| Anti-Tuj1(B-Tubulin III) | Sigma | T8660 |
| Anti-Ki-67 | eBioscience | |
| Anti-CD24a | eBioscience | M1/69 |
| Anti-CD56 | BD Biosciences | 809220 |
| Anti-SETD2 | AbClonal | A3194 |
| H3K36me3 | Abcam | Ab9050 |
| H3K79me2 | Millipore | 04-835 |
| H2BK120Ub | Medimabs | MM-0029-P |
| Histone H1 | Millipore | AE04 |
| NEUROG1 | Santa Cruz | Sc-100332 |
| Lamin B1 | Santa Cruz | Sc-374015 |
| HA | Abcam | Ab9110 |
| Streptavidin-HRP | pierce | 21130 |
| Total Pol2 | MABI0601 | MBL |
| H3K4me3 | Abcam | Ab8580 |
| H4-panacetyl | ActiveMotif | B_2687872 |
| H3K27me3 | Diagenode | C15410195 |
| **Oligonucleotides and other sequence-based reagents** | | |
| Primers | This study | Dataset EV4 |
| **Chemicals, enzymes, and other reagents** | | |
| DMEM | Thermo | 61965059 |

| Reagent/resource | Reference or source | Identifier or catalog number |
|---|---|---|
| FCS | Gibco | |
| Non-essential AA | Thermo | 11140035 |
| Glutamax | Thermo | 35050061 |
| β-mercaptoethanol | Thermo | 31350010 |
| LIpfectamine 3000 | Thermo | L3000015 |
| OptiMEM | Thermo | 31985070 |
| Zeocin | Invitrogen | R25001 |
| dTAG13 | Sigma | SML2601 |
| EPZ-719 | Sigma | SML3949 |
| DMSO | VWR | A3672 |
| Foxp3/Transcription Factor Staining Buffer | Thermo | 00-5523-00 |
| DAPI | Sigma | H-2000-2 |
| LIVE/DEAD Fixable Near-IR Dead Cell Stain | Invitrogen | L34975 |
| Peroxidase IHC Detection Kit | Thermo | 36000 |
| RNeasy Plus mini | Qiagen | 74134 |
| SuperScript III First-Strand Synthesis | Invitrogen | 18080051 |
| RnaseH | NEB | M0297S |
| KAPA SYBR FAST qPCR Kit | Sigma | |
| **Software** | | |
| Fiji | | version 21.07/1.54p |
| FlowJO | Tree Star | Version 10.7 |
| nf-core/rnaseq | https://github.com/nf-core/rnaseq | Version 3.12.0 |
| trimgalore | https://github.com/FelixKrueger/TrimGalore | |
| STAR | Dobin et al, 2013 | |
| salmon | https://github.com/COMBINE-lab/salmon | Version 1.10.1 |
| EdgeR | Robinson et al, 2010 | Version 4.6.3 |
| FGSEA | Korotkevich et al, 2016 | |
| ISMARA | Balwierz et al, 2014 | https://ismara.unibas.ch/mara/ |
| rMATS-turbo | Wang et al, 2024 | Version 4.0.2 |
| cutadapt | Martin, 2011 | Version 4.9 |
| STARsolo | Kaminow et al, 2021 | Version 2.7.11 |
| scanpy | Wolf et al, 2018 | Version 1.10 |
| miloDE | Missarova et al, 2024 | Version 4.5 |
| Bowtie2 | https://github.com/BenLangmead/bowtie2 | Version 2.5.1 |
| SAMtools | https://www.htslib.org | Version 1.20 |
| picard | https://broadinstitute.github.io/picard/ | Version 3.1.1 |

| Reagent/resource | Reference or source | Identifier or catalog number |
|---|---|---|
| Deeptools | https://deeptools.readthedocs.io/en/latest/ | Version 3.5.5 |
| QuasR | Gaidatzis et al, 2015 | Version 4.5 |
| Enrichr | Kuleshov et al, 2016 | https://maayanlab.cloud/Enrichr/ |
| **Other** | | |
| NEB Next Poly(A) mRNA Magnetic Isolation Module | NEB | |
| NEB Next UltraTM II Directional RNA Library Prep Kit for Illumina | NEB | |
| Nextera DNA Library Prep Kit | Illumina | |
| AmpureXP beads | Beckman Coulter | |
| MinElute PCR purification columns | Qiagen | |
| TapeStation2200 | Agilent | |
| Illumina HiSeq 4000 | Illumina | |
| Illumina NovaSeq | Illumina | |
| LSM 700 | Zeiss | |
| Plan-Apochromat 63×1.4 | Zeiss | |
| FACSFortessa | BD Biosciences | |
| QuantStudio 5 Real-Time PCR System | Applied Biosciences | |

## Cell line generation, cell culture, and neuronal differentiation

Mouse embryonic stem cells (HA36CB1) were cultured on 0.2% gelatine-coated dishes in DMEM (Invitrogen), supplemented with 15% fetal calf serum (Invitrogen), 1× non-essential amino acids (Invitrogen), 1× Glutamax (Invitrogen), homemade leukemia inhibitory factor (LIF), and 0.001% β-mercaptoethanol (Invitrogen) at 37 °C and 7% $CO_2$. ESC lines were differentiated as previously described (Bibel et al, 2004), except that no feeder cells were used. Microscopy images were taken at ×100 magnification. Cell count assays were performed using live/dead stain and TC20™ Automated Cell Counter (BioRad).

*Setd2*-KO in *Dnmt*-triple KO mouse ESC background was generated using CRISPR/Cas9. The Cas9 sgRNA sequences, as previously described (Baubec et al, 2015), were cloned into the pX330-U6-Chimeric_BB-CBh-hSpCas9 (Addgene 42230). Transfections together with pRR-Puro recombination reporter (Addgene 65853) were conducted using Lipofectamine 3000 reagent (L3000015, Thermo Fisher Scientific) at a 2:1 Lipofectamine/DNA ratio in OptiMEM (31985070, Thermo Fisher Scientific). Thirty-six hours later, cells were selected with 2 µg/ml puromycin for another 36 h. ESCs containing a homozygous CGT to CAC in-frame mutation at position R1599H of *Setd2* were generated using CRISPR/Cas9. Here, donor plasmids for homologous repair, containing 1 kb homologous sequences upstream and downstream of the cut site and a mutated PAM site, were co-transfected in a

1:2.5 (sgRNA:donor) ratio together with sgRNA and the pRR-Puro reporter using lipofection. The R1599H mutation was confirmed by PCR and Sanger sequencing, as well as immunoblotting for H3K36me3 levels.

*Setd2*-KO ESCs with inducible expression of Neurogenin 1 and 2 (iNgn1/2) were generated as described previously with adaptations (Busskamp et al, 2014). A doxycycline-inducible rtTA3 system (Addgene 61472) was randomly integrated using 20 μg of linearized plasmid with bleomycin resistance. Cells were treated with 200 μg/ml Zeocin (InvivoGen). The TetON-inducible Ngn2-2A-Ngn1 ESCs in the *Setd2*-KO background were then obtained by recombinase-mediated cassette exchange (RMCE) (Baubec et al, 2013) with an expression plasmid for CRE recombinase in a 1:0.6 DNA ratio. Similarly, shRNA-inducible ESCs were generated using a TetON-inducible system for a shRNA against *Setd2*, expressed from the RMCE site. Positive clones were confirmed by Sanger sequencing, RT-qPCR, and immunoblotting. Oligo sequences available in Dataset EV4. Inducible shRNA-mediated knockdown as well as iNgn1/2 cell lines were treated with 1 μg/ml doxycycline (Sigma-Aldrich). The Setd2-FKBP dTAG mESC lines were generated and validated as described in (Molenaar et al, 2022), and dTag13 (Sigma) treatments were conducted at 0.5 μM concentrations for different durations as described in the text. Treatments with the SETD2 inhibitor EPZ-719 (Sigma) were performed at 0.5 μM as described (Lampe et al, 2021). All cell lines used in this study tested negative for mycoplasma contamination.

## Immunofluorescence microscopy

mESCs seeded onto 0.2% gelatin-coated 12-well removable chambers (Ibidi, 81201) were incubated with 50 μM Biotin for 1 h. Immunofluorescence was performed according to (Schmolka et al, 2023), using the following primary antibodies: anti-Biotin (Sigma-Aldrich—3737373, 1:500) and anti RNAPII-S2P (Abcam—ab5095, 1:1000), diluted in 1% BSA blocking buffer. For TN immunofluorescence, NPCs were seeded on acid-washed coverslips (0.13-0.16 mm—Ted Pella, #26020), coated with poly-L-ornithine-laminin and immunofluorescence was performed as above using Anti-B-Tubulin III (Tuj1), (Sigma T8660, 1:2000). Images were acquired with a confocal laser scanning microscope Zeiss LSM 700 (Biology Image Center, Utrecht University, Netherlands) with a Plan-Apochromat 63×1.4 NA oil immersion objective (Carl Zeiss, Germany) using the 405, 488 and 568-nm lasers. Images were processed with Fiji (version 21.07/1.54p). Colocalization coefficients (Mander's correlation), calculated from pixel intensities within regions of interest generated from DAPI-stained nuclear segmentation, were obtained using the coloc2 plugin from Fiji. Where necessary, cells were equally adjusted for brightness and contrast.

## Surface marker and cell cycle analysis using flow cytometry

Single-cell suspensions were obtained through trypsinization and filtered through 40-um cell strainers (BD Biosciences). For cell cycle and Ki-67 measurements, single-cell suspensions were fixed and permeabilised for 30 min at 4 °C with Foxp3/Transcription Factor Staining Buffer set (eBioscience). Anti-Ki-67 or isotype control (eBioscience) was added and incubated for 45 min at RT in permeabilisation buffer. DAPI (5 μg/ml, Sigma) was added as a fluorescent DNA stain 5 min prior to FACS measurements and

incubated at RT in the dark. For CD24 and CD56 measurements in neuronal progenitors, single-cell suspensions were obtained from neuronal progenitors after 8 days of differentiation. Cells were incubated for 30 min at 4 °C with a saturating concentration of anti-CD24a monoclonal antibody (1:200) (eBioscience, Clone M1/69) and anti-CD56 monoclonal antibody (1:200) (BD Biosciences, clone 809220). LIVE/DEAD Fixable Near-IR Dead Cell Stain (L34975, Invitrogen) was used to discriminate cell viability. Samples were acquired using a FACSFortessa (BD Biosciences), and data were analyzed using FlowJo software (version 10.7, Tree Star).

## Protein extraction and immunoblotting

Crude nuclear extracts from cells were obtained as described in ref. (Manzo et al, 2017), histones were acid-extracted according to ref. (Villaseñor et al, 2020). Membranes were blocked with 5% milk in TBS with 0.1% Tween20 and incubated with primary antibodies against SETD2 (1:1000, A3194, ABclonal, LOT 0071240201), H3K36me3 (1:5000, ab9050, Abcam, LOT GR3210075-1), H3K79me2 (1:2000, 04-835, Milipore, clone NL-59), H2BK120ub (1:2000, MM-0029-P, Medimabs, clone NRO3), anti-histone H1 (1:5000, AE04, Millipore, LOT 3087175), anti-NEUROG1 (1:1000, sc-100332, Santa Cruz Biotechnology, LOT F1119), anti-LAMIN B1 (1:1000, sc-374015, Santa Cruz Biotechnology, LOT J3019), anti-HA (1:1000, ab9110, Abcam), Streptavidin-HRP (1:20000, Pierce, clone 21130), overnight at 4 °C. Protein detection was facilitated using species-specific antibodies conjugated to horseradish peroxidase and Pierce® Peroxidase IHC Detection Kit (Thermo Scientific).

## RNA isolation, cDNA synthesis, and RT-qPCR

RNA was isolated from mouse ESCs, cellular aggregates, NPCs, and terminal neurons using the RNeasy Plus mini kit (Qiagen). Coding DNA was synthesized from 2 μg isolated RNA with SuperScript III First-Strand Synthesis (Invitrogen) for 60 min at 50 °C using Oligo(dT) (Thermo Fisher), followed by heat-inactivation for 10 min at 85 °C. Residual RNA was digested with two units RNaseH (NEB) for 20 min at 37 °C. Target sequences were quantified by real-time qPCR analysis using a KAPA SYBR FAST qPCR Kit (Sigma-Aldrich) on a QuantStudio 5 Real-Time PCR System (Applied Biosystems). Comparative quantification (ddCt) was used to determine transcript levels relative to the housekeeping gene *Hprt*. Each sample was run in at least technical triplicate. Oligo sequences are available in Dataset EV4.

## PolyA RNA-sequencing and differential gene expression analysis

Total RNA was isolated from NPCs using the RNeasy Plus mini kit (Qiagen). RNA integrity was measured using a model 2100 Bioanalyzer (Agilent). PolyA-tailed mRNAs were isolated and enriched using NEB Next Poly(A) mRNA Magnetic Isolation Module according to the manufacturer's instructions. Libraries for 1 μg mRNA were prepared using NEB Next UltraTM II Directional RNA Library Prep Kit for Illumina. Sequencing of library pools and read processing were performed on Illumina HiSeq4000 according to Illumina standards, with 125-bp single-end sequencing. Quality

control, read mapping, and quantification were performed using nf-core/rnaseq v3.12.0. Reads were adapter- and quality-trimmed with trimgalore and aligned to the *Mus musculus* reference genome (GRCm39/mm39) using STAR (Dobin et al, 2013). Transcript and gene-level quantification were performed with Salmon (v1.10.1), and gene annotation was based on GENCODE release M35 (gencode.vM35.annotation.gtf). Differential gene expression analysis was performed using the edgeR package (Robinson et al, 2010). Genes with low expression across all samples were filtered with filterByExpr. Library sizes were normalized using the trimmed mean of M values (TMM) method, and differential expression was assessed with the quasi-likelihood framework (qlmQLFit) followed by glmTreat, testing for changes with |log2 fold change| >1 at a false discovery rate (FDR) < 0.05. Gene ontology enrichment analysis on differentially expressed genes was performed using the goana() function in edgeR. GSEA analysis was performed using FGSEA (Korotkevich et al, 2016), while collapsing similar pathways for better representation. Motif response analysis was performed using the ISMARA online tool (Balwierz et al, 2014). Figures, including MA plots, violin plots of gene sizes, and GSEA bar plots, were generated using ggplot2.

## Alternative splicing analysis

Differential alternative splicing events were analyzed using rMATS-turbo (v4.0.2) (Wang et al, 2024), starting from RNA-seq BAM files. rMATS detects and quantifies five major classes of alternative splicing events (skipped exons, alternative 5′ splice sites, alternative 3′ splice sites, mutually exclusive exons, and retained introns) by integrating junction and exon body reads. For downstream filtering, events were required to meet the following criteria: a minimum read coverage of 10 and a percent spliced-in (PSI) between 0.05 and 0.95. Significant events were defined as those with FDR < 0.01 and an absolute $\Delta$PSI $\geq$ 0.05.

## Single-cell RNA-sequencing and data analysis

scRNA-seq was performed in 384-well plates according to the sortSeq protocol as described before (Muraro et al, 2016). Paired-end .fastq files were obtained for each sample, where Read-1 contains the cell barcodes in celseq2 format (6-base UMI; 8-base barcode), and Read-2 contains the 3′-mapping reads from the transcript. The raw reads were trimmed for quality with cutadapt (v4.9) (Martin, 2011), mapped to mm39/GRCm39 genome using STARsolo workflow (v2.7.11) (Kaminow et al, 2021), to yield gene-level counts based on the genomic annotation from gencode (v35). The mapping and counting workflow is available here: https://github.com/bhardwaj-lab/snakepit/blob/master/map_scRNAseq.Snakefile (v.1.0). Downstream analysis was performed via scanpy (v1.10) (Wolf et al, 2018). Cells were filtered based on the following parameters: total_counts <1000, protein_coding fraction >0.7, n_genes_by_counts <=10,000, pct_counts_in_top_100_genes <80. Counts were normalized to 10,000 (excluding highly expressed genes) and log-transformed. PCA was performed on 3000 highly variable genes, 15 nearest neighbors were used to create a neighborhood graph, and Leiden clustering was performed on the graph. UMAP projection was performed on the PAGA graph using parameters: min_dist=0.5, spread=5, init_pos = 'paga'. Marker genes were detected per cluster using the 't-test_overestim_var'

method, and the clusters were manually annotated into five groups based on the markers. For differential expression analysis, we used the PCA output to create a 2nd order KNN graph ($k = 20$) and performed DE analysis between wild-type (WT) and knockout (KO) cells per neighborhood using miloDE in R (v4.5) (Missarova et al, 2024), which uses edgeR (v4.6.3) to compare the aggregated WT and KO counts per neighborhood. The corrected $P$ values across all genes ($P < 0.05$) were used to identify DE genes, which were then used for plotting.

## Chromatin immunoprecipitation (ChIP) and ATAC-sequencing and read processing

Histone ChIP experiments and sequencing were performed as previously described (Villaseñor et al, 2020). Here, 100 µg chromatin were incubated with 5 µg of either H3K36me3 (ab9050, Abcam), total Pol II (MABI0601, MBL), Ser2-P RNA Pol II (ab5095, Abcam), H3K4me3 (ab8580, Abcam), H4-panacetyl (B_2687872, ActiveMotif), or H3K27me3 (C15410195, diagenode) antibody. Sequencing of library pools was performed on Illumina NovaSeq according to Illumina standards, with 150-bp single-end sequencing. The ATAC-seq reaction was performed with 50,000 cells as previously described, with minor adjustments of the protocol, using the Nextera DNA Library Prep Kit (Illumina) together with the barcoded primers from the Nextera Index Kit (Illumina). In brief, an additional size selection step was performed after the first five cycles of library amplification. For this, the PCR reaction was incubated with 0.6× volume of Ampure XP beads (Beckman Coulter) for 5 min to allow binding of high-molecular-weight fragments. Beads containing long DNA fragments were separated on a magnet, and the supernatant containing only small DNA fragments below roughly 800 bp was cleaned up using MinElute PCR purification columns (Qiagen). All libraries were amplified for 12 cycles in total, visualized and quantified with a TapeStation2200 (Agilent), and sequenced as mentioned above.

Library demultiplexing was performed following Illumina standards. Samples were filtered for low-quality reads as well as adapter sequences using trimgalore. All files were aligned to the reference mouse genome assembly GRCm39 (mm39) using Bowtie2 (v2.5.1). Multiple-matching reads were eliminated using SAMtools (v1.20). Duplicated reads were removed by Picard (v3.1.1). DeepTools (v3.5.5) was used for read depth normalization and visualization. Wiggle tracks were obtained with QuasR and visualized with the UCSC genome browser.

## ChIP-seq analysis

For promoters/gene body analyses, sequencing depth-normalized read counts were computed in promoter regions (2 kb around TSS) and gene bodies (+ 2 kb downstream from TSS). For genome-wide cross-correlations, the mm39 genome was tiled into 1-kb windows, and per-tile normalized read counts were obtained. Pairwise Spearman correlations were visualized as heatmaps. For the calculation of the pausing index, ChIP-seq signals of total RNA Pol II were used. The pausing index was calculated as the ratio of normalized counts in the promoter window ($-50$ bp to $+300$ bp around the TSS) over the early gene body ($+ 650$ bp to $+1000$ bp from TSS) for genes with length greater than 2 kb. For quantification of H3K36me3 signal over the spliced regions, skipped exon,

retained intron, and mutually exclusive exon events were used. H3K36me3 signal was normalized for sequencing depth and region length, with the spliced region defined as the genomic interval spanning from the start of the upstream constitutive exon to the end of the downstream constitutive exon.

## ChromID and label-free MS data acquisition and analysis

ChromID samples were prepared as previously described in ref. (Villaseñor et al, 2020). In brief, nuclear extraction of SRI-TurboID NPCs of wild-type and Setd2-KO background was performed after biotin incubation for 12 h, followed by affinity purification through streptavidin beads, high-stringency washes, and on-bead digestion. MS data acquisition was performed as described in ref. (Villaseñor et al, 2020) using randomization. In brief, samples (quadruplicates per condition) were cleaned up by C18 StageTips. Peptides were detected by data-dependent acquisition via mass spectrometry. Proteins were identified and quantified from the raw acquisition data as well as processed using MaxQuant (Cox and Mann, 2008). The mouse reference proteome (UniProtKB/Swiss-Prot) version 2018_12 combined with manually annotated contaminant proteins was searched with protein and peptide false discovery rate (FDR) values set to 1%, and the match-between-runs algorithm was enabled. Statistical analysis was conducted using Proteus (Gierlinski et al, 2018). LFQ intensity values were log2-transformed, and outlier samples were determined based on low peptide or protein counts. Subsequently, Proteus' limma-wrapper (Ritchie et al, 2015) was used to determine potential interactors in respective contrasts (bait vs. control of the same genetic background) with an FDR of 0.05 as a significance threshold. For gene ontology term analysis, differentially enriched proteins of bait conditions were combined and parsed to Enrichr (Kuleshov et al, 2016).

## Data availability

High-throughput sequencing data obtained from ChIP-seq, ATAC-seq, RNA-seq, and single-cell RNA-seq studies were deposited to NCBI Gene Expression Omnibus under the following accessions: GSE278955 (RNA-seq), GSE278956 (ChIP-seq), GSE278957 (ATAC-seq), GSE321694 (scRNA-seq). Scripts used to analyze genomics and proteomics data were deposited to https://github.com/BaubecLab/Setd2.

The source data of this paper are collected in the following database record: biostudies:S-SCDT-10_1038-S44318-026-00768-2.

## Peer review information

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

## Acknowledgements

We thank members of the Baubec laboratory for their input and criticism. Furthermore, we thank members of the Functional Genomics Centre Zurich for their genomics and proteomics support. This work was supported by the Swiss National Science Foundation through SNSF Professorship (183722) and SNSF Sinergia (180354), by the European Research Council (865094 - ChromatinLEGO - ERC-2019-COG), and the EMBO Young Investigator program. CA acknowledges support from the UZH Candoc Grant, and NS acknowledges support from the EMBO postdoctoral fellowship and SNSF Ambizione grant (186012).

## Author contributions

**Christina Ambrosi**: Formal analysis; Investigation; Writing—original draft. **Ramon Pfaendler**: Investigation; Methodology. **Kristeli Eleftheriou**: Data curation; Formal analysis; Methodology. **Stefan Butz**: Methodology. **Davide Recchia**: Investigation; Methodology. **Xue Bao**: Investigation; Methodology. **Richard Cardoso da Silva**: Data curation; Formal analysis; Investigation; Methodology. **Niklas Kupfer**: Investigation; Methodology. **Ilse M Lagerwaard**: Investigation. **Hanneke Vlaming**: Formal analysis; Supervision; Investigation. **Nina Schmolka**: Formal analysis; Supervision; Investigation; Methodology. **Vivek Bhardwaj**: Data curation; Formal analysis. **Tuncay Baubec**: Conceptualization; Supervision; Funding acquisition; Investigation; Writing—original draft; Writing—review and editing.

Source data underlying figure panels in this paper may have individual authorship assigned. Where available, figure panel/source data authorship is listed in the following database record: biostudies:S-SCDT-10_1038-S44318-026-00768-2.

## Disclosure and competing interests statement

The authors declare no competing interests.

