## [Peer Review File · The EMBO Journal]

The H3K36me3 methyltransferase SETD2 contributes to PAF1C interactions with RNA Pol II and is required for neuronal differentiation

Christina Ambrosi, Ramon Pfaendler, Kristeli Eleftheriou, Stefan Butz, Davide Recchia, Xue Bao, Richard Cardoso da Silva, Niklas Kupfer, Ilse Lagerwaard, Hanneke Vlaming, Nina Schmolka, Vivek Bhardwaj, and Tuncay Baubec

Corresponding author(s): Tuncay Baubec (t.baubec@uu.nl)

Review Timeline:

Submission Date:	27th Feb 25
Editorial Decision:	14th Mar 25
Revision Received:	20th Sep 25
Editorial Decision:	11th Dec 25
Revision Received:	23rd Jan 26
Editorial Decision:	26th Feb 26
Revision Received:	27th Feb 26
Accepted:	11th Mar 26

Editor: Cornelius Schneider

Transaction Report:

Reviewer #1 (Remarks to the Author):

Ambrosi et al. investigate the role of SETD2 in neurodevelopment, presenting intriguing findings about its function in neuronal differentiation. The authors demonstrate that Setd2 KO, similar to Dnmts or PRC2 KOs, impairs cell identity acquisition while being viable in stem cells. Using mouse ESCs and a novel quadruple KO combining Setd2 and Dnmts TKO, they show that SETD2 is essential for neuronal precursor cell establishment but dispensable for neuronal differentiation, following the Bibel et al. protocol (PMID: 17546008). In contrast, Dnmts TKO and Eed KO failed to differentiate into neurons, consistent with previous reports for Suz12 KO (PMID: 38452766).

A key finding is that SETD2's role in neurodevelopment may be independent of its enzymatic activity (H3K36me3 deposition). This is supported by the lack of correlation between gene expression and H3K36me3 (Fig. 3C) and the ability of catalytically dead SETD2 lines to differentiate into mature neurons. The study integrates proteomics and genome engineering to explore these mechanisms.

While the paper is well-written and presents a clear hypothesis, some results look preliminary and require stronger experimental support. Additionally, the RNA-seq and ChIP-seq analyses are superficial and should be expanded to strengthen the findings.

Major Point:

1) H3K36me3 and Alternative Splicing (Fig. 3C):

The authors report no correlation between differentially expressed genes and K36me3 deposition in Setd2 KO day 8 NPCs. It would be valuable to analyze changes in alternative splicing (AS) of exons and introns where K36me3 levels are reduced. Single-end 150 bp RNA-seq should suffice to estimate AS changes.

This is particularly relevant because Neuronal tissues exhibit high AS diversity; SETD2 interacts with splicing factors like hnRNP L (PMID: 33664260); K36me3 regulates AS of splicing factors (e.g., SRSF11; PMID: 34480866); RNA Pol II elongation affects splicing decisions (PMID: 30988016).

2) RNA Pol II ChIP-seq and Pausing Index:

TurboID-MS suggests reduced RNA Pol II elongation in SETD2 KO. The authors performed RNA Pol II ChIP-seq, which can serve to access the genome-wide pausing index at differentially expressed genes. If SETD2 affects Pol II elongation via PAF1C, changes in the pausing ratio (promoter vs. early gene body) would be expected.

3) Ngn1/2 Overexpression and Transcriptional Initiation:

The authors speculate that Ngn1/2 overexpression boosts PAF1C engagement with Pol II at neuronal genes, enforcing transcriptional initiation. However, Ngn1/2 primarily acts on promoters and enhancers. The claim requires stronger evidence, as increased post-mitotic differentiation in NPC cultures could occur independently of PAF1C engagement.

4) Phenotypic Characterization of SETD2 KO Neurons (Fig. 1, Fig. 5):

Differentiation estimates are based on bright-field microscopy images, which lack quantitative detail. The authors should improve phenotypic characterization with immunofluorescence staining for neuronal markers. For defective cultures, how is neuronal survival estimated (Figs. 1B, C; 3B)? It's unclear if SETD2 KO leads to widespread cell death or a neurite growth defect. Quantifying these phenotypes during neuronal maturation, as previously reported (PMID: 28106138), would clarify the impact of SETD2 KO.

5) The authors do not comment on the contribution to H3K36 methylation from NSD1-3 and ASHL1. It would be helpful to clarify in the introduction which methylation states are specifically deposited by SETD2.

Minor Points:

1) Why did the authors use the mm9 genome assembly instead of the more updated mm10 or mm39?

2) In Figure 4d, why weren't all proteins identified in Figure 4b included in the network reconstruction? Additionally, the meaning of "pooled background set" (line 189) is unclear.

Reviewer #2 (Remarks to the Author):

In this manuscript, Ambrosi et al., investigate the role of histone and DNA methyltransferases in neuronal differentiation potential of mESCs, focusing on SETD2, a H3K36me3 methyltransferase. The authors demonstrate that SETD2 is important early on during the establishment of neuronal gene expression pattern but is dispensable once cells have differentiated. Further, they find that this is largely a non-catalytic function of SETD2, since the catalytic-dead enzyme was able to rescue differentiation. Mechanistically, using BioID experiments, they claim that SETD2 facilitates RNAPII interaction with the PAF1 complex, which is important for SETD2 mediated establishment of neuronal gene expression programs.

While the question being addressed and the initial observation that SETD2's catalytic activity is dispensable for differentiation are both interesting, the manuscript contains significant deficiencies that render it unsuitable for publication at this journal.

Major points:

1. The evidence for proposed mechanism is very rudimentary and technically problematic. First, using SRI domain fused to TurboID as a proxy to map RNAPII interactome, rather than tagging any RNAPII subunit, is a very surprising approach. Why didn't the authors simply tag one of the RNAPII subunits, like Rbp3? While it is known that SRI domain binds Ser2phos-Rbp1, one can't be sure of the localization dynamics of the overexpressed SRI-TurboID fusion. There is also evidence that the SRI can associate with DNA (PMID: 25925577).
2. Second, the authors' claim that SETD2 promotes RNAPII interaction with PAF1 complex is not substantiated by the data. BioID mass-spec experiments are prone to a lot of variations and the differences in PAF1 complex members are very minimal and could be within the margin of error. Additionally, there is no validation of the BioID experiments using orthogonal approaches, such as co-IP. Without validation of BioID experiments, it is very hard to interpret any of the data.
3. Third, the authors' claim that the observed phenotype in Setd2-KO cells is due to reduced PAF1 complex with RNAPII is also not supported by their data. The attempt to rescue Setd2-KO cells by overexpressing neuronal transcription factors doesn't prove that SETD2 acts through promoting PAF1 association with RNAPII. Overexpression of such master TFs could potentially override any number of molecular manipulations and it is almost impossible to categorically point to one specific mechanism.
4. The molecular characterization of differentiation is weak. The only read out the authors use for differentiation is the percentage of cells that survive the differentiation protocol. However, there is no attempt to properly characterize whether the surviving cells exhibit properties of properly differentiated neuronal cells, except for a cursory attempt in the last figure.
5. While the CRISPR mediated R1599H point mutant is reasonable, the authors should have also used the SETD2 inhibitor (EPZ-719) to orthogonally test if SETD2's catalytic activity is required for differentiation.
6. The ChIPseq data in Fig 3a are very noisy and also not well analyzed. There's no global analysis of all the data except for H3K36me3 and H4-panAc

Overall, the manuscript in its current form is in a very elementary stage and many of the author's claims are not substantiated by the data and therefore it would take an enormous amount of work to meet the standards of NSMB for publication.

Reviewer #3 (Remarks to the Author):

In the manuscript "SETD2 promotes PAF1C interactions with the elongating RNA Pol II and is required for neuronal differentiation" Ambrosi et al. set out to compare the effects of depleting key

chromatin factors by CRISPR/Cas9 in isogenic mES cell lines on neuronal differentiation. I consider this unification effort important, as often in the chromatin field work is focused on one factor and compared to results obtained using a model generated in a different lab using another background. The neuronal differentiation experiments showed, that the various chromatin regulators tested (EED, DNMT1, DNMT3A, DNMT3B, and SETD2) were required at different stages of cellular differentiation. While EED and DNMTs were required early in the differentiation protocol (Eed-KO cells did not survive exit from pluripotency and Dnmt-TKO cells failed differentiation at the neural progenitor stage), SetD2, the enzyme responsible for H3K36me3, was important during late stages of differentiation, for the establishment of proper neuronal gene expression. Importantly, the authors reproduced their results in two independent knock-out clones and one constitutive Setd2 knock-down cell line. This is in line with mouse KO model which previously showed that Setd2^{-/-} is embryonically lethal only at day E10.5, leading to growth defects including forebrain hypoplasia and unclosed neural tubes (Hu, M. et al. PNAS 2010). Further on, the authors demonstrated in the current manuscript using SETD2 depletion in a dox-inducible system that SETD2 was dispensable once the cells have fully differentiated. The authors then checked whether the lysine methyltransferase activity of SETD2 is important for this function, by generating a methyltransferase catalytic-dead mutant (R1599H) knock-in. Surprisingly, this had only a mild effect, implying that the function of SETD2 in neuronal differentiation is mostly independent of its catalytic activity. The authors employed then an elegant approach, they established in their own lab previously, ChromID, to measure the protein interaction network of elongating RNA Pol II. They fused the SRI domain of SETD2, which binds the serine-2-phosphorylated CTD of RNA Pol II, to the promiscuous biotin ligase TurboID. This enabled them identification of proteins associated with the elongating RNA Pol II by proximity biotinylation. This revealed that multiple subunit of the transcription elongation complex PAF1 were less associated to elongating RNA Pol II. The authors concluded that they identified a novel role for SETD2 in mediating interactions between the PAF1 complex and the elongating RNA Pol II, which is required to ensure optimal transcriptional processivity of neuronal genes, and independent of its catalytic activity.

I find the presented approaches valid and quality of data high. The conclusions are well justified by the data. Besides the important insights obtained directly from the study, the obtained isogenic cell lines will be a valuable resource for the chromatin community

My feeling is that the results presented are of immediate interest to many people in the transcription and chromatin fields, who are a significant fraction of readers of NSMB, and potentially also to biomedical researchers working on epigenetic therapies. I think the length of the text is appropriate to describe the results. I would only suggest to expand the discussion (slightly) and methods (more) sections, see points below.

Overall, I recommend this manuscript for publication in NSMB.

Nevertheless, there are several small but important issues that need to be revised in my view.

1. Most importantly – ChromID methodological description.

I find the modular building blocks approach of the ChromID method very elegant. In particular, analysing RNA Pol II interactome by proximity biotinylation has many advantages over more traditional immunoprecipitation-based methods. However, as this is still a novel method, and not standard in the field, it should be much better described methodologically, in particular in respect to the analysis. Ideally a script should be made available to allow full reproducibility of the results obtained. Minimally, much more details of the approach, parameters used etc.

2. Similarly, a script for the NGS data analysis in github or zenodo would be helpful for reproducibility.

3. Since the authors performed RNA-seq in wt and SETD2 KO conditions, did they see any indication of missplicing? This appears to occur in some models in literature but not all.

4. I would like to see more information/discussion about non-PAF1 factors found in ChromID. Eg, KMT2E appears the most depleted (fig S8B) in SETD2 KO, could the authors comment on this? Figure 4B – does the high SETD2 values in the SETD2 KO come from the SRI autobiotinylation? Please indicate this in manuscript.

5. Figure S6f – in the authors' opinion, does the remaining H3K36me3 signal in SETD2 KO correspond more likely to residual H3K36me3 or cross-reactivity of the antibody with K36me2? I would suggest to openly mention this in the manuscript.

6. Fig 3b Authors say “modest correlation”, but indicate no p-values. I assume the number on the graph is a correlation coefficient, but this is not explained in the legend.

Reviewer #4 (Remarks to the Author):

Ambrosi et al explore the role of the H3K36 trimethyltransferase Setd2 during exit from pluripotency and differentiation into neural progenitor cells (NPCs) and neurons using mouse ESCs. They find that while Setd2 is dispensable for pluripotency exit and NPC generation, it is required for the establishment but not maintenance of neurons. They further show that this phenotype is driven by a non-catalytic function of Setd2. Proximity labeling combined with mass spec suggests that Setd2's role in differentiation may be mediated by the association of Pol2Ser2 with the Paf1 complex to safeguard transcriptional processivity of neuronal targets. Consistent with this idea, overexpression of a neuronal TF partially rescues the differentiation defect of Setd2^{-/-} cells.

Overall, this is a well-performed study that provides insights into the non-catalytic role of Setd2 in neuronal differentiation, which appears to be mediated at least in part via the Paf1 complex implicated in transcriptional elongation. However, the authors do not directly show that there is a defect in transcriptional processivity. Moreover, several of the presented results are confirmatory of previous observations, such as the differentiation defects of Eed, Dnmt TKO cells and the lack of a strong correlation between loss of H3K36me3 in Setd2 KO cells and gene expression changes. The manuscript in its current form may therefore be a better fit for a more specialized journal.

Specific points:

1. Survival is not a specific assay to infer neuronal differentiation defects. The authors need to perform IF for appropriate markers in control and all mutant cells to conclusively show that there is indeed a differentiation defect.
2. The authors' conclusion that Setd2 is not required for neuronal maintenance is interesting. However, they need to confirm that Setd2 is still efficiently depleted by Western blot analysis (i.e. is the hairpin vector still expressed?) at the end point of differentiation to rule out silencing of the vector.
3. The authors should perform RNA-Seq of the catalytically dead Setd2 cells before and after differentiation and compare with WT and Setd2 KO cells to better appreciate how the catalytic and non-catalytic roles of Setd2 affect transcriptional output globally.
4. The authors should consider doing Gro-Seq or Pro-Seq in WT and mutant cells to directly test whether there are defects in transcriptional processivity, as proposed by their model.
5. The authors need to include a WT minus Dox control in their rescue experiment in Fig. 5c to assess whether candidate gene expression, particularly Ngn1/2, is different in Setd2 KO cells. If Ngn1/2 levels are down in Setd2 KO cells, this could be a more trivial explanation to explain the differentiation defect and the authors' ability to rescue the phenotype via Ngn1/2 overexpression. Do the authors expect that other neuronal TFs have a similar effect, i.e. is there some specificity to Ngn1/2?
6. Statistical analyses are missing in several places, such as Figs. 1b, 3d, 5c, etc. and need to be provided.
7. Zhang et al previously showed that Setd2 is required for ESC differentiation towards primitive endoderm (PMID: 25242323). The authors should discuss this finding in the context of their results.
8. Do the different colors in Fig. 4d have any particular meaning?

Dear Prof. Baubec,

Thank you for submitting your manuscript for consideration by the EMBO Journal and for the productive discussions during our meeting. Based on your revision plans discussed during this meeting we invite you to submit a revised version of the manuscript.

Thank you for the opportunity to consider your work for publication. I look forward to your revision.

Yours sincerely,

Cornelius Schneider, PhD
Editor
The EMBO Journal
c.schneider@embojournal.org

We realize that it is difficult to revise to a specific deadline. In the interest of protecting the conceptual advance provided by the work, we recommend a revision within 3 months (12th Jun 2025). Please discuss the revision progress ahead of this time with the editor if you require more time to complete the revisions. Use the link below to submit your revision:

Response to Reviewer's reports

Reviewer #1 (Remarks to the Author):

Ambrosi et al. investigate the role of SETD2 in neurodevelopment, presenting intriguing findings about its function in neuronal differentiation. The authors demonstrate that *Setd2* KO, similar to Dnmts or PRC2 KOs, impairs cell identity acquisition while being viable in stem cells. Using mouse ESCs and a novel quadruple KO combining *Setd2* and Dnmts TKO, they show that SETD2 is essential for neuronal precursor cell establishment but dispensable for neuronal differentiation, following the Bibel et al. protocol (PMID: 17546008). In contrast, Dnmts TKO and Eed KO failed to differentiate into neurons, consistent with previous reports for Suz12 KO (PMID: 38452766).

A key finding is that SETD2's role in neurodevelopment may be independent of its enzymatic activity (H3K36me3 deposition). This is supported by the lack of correlation between gene expression and H3K36me3 (Fig. 3C) and the ability of catalytically dead SETD2 lines to differentiate into mature neurons. The study integrates proteomics and genome engineering to explore these mechanisms.

While the paper is well-written and presents a clear hypothesis, some results look preliminary and require stronger experimental support. Additionally, the RNA-seq and ChIP-seq analyses are superficial and should be expanded to strengthen the findings.

Response #01: We have initially performed extensive analysis of RNA- and ChIP-seq datasets related to this study. However, in the previous version, we decided to not include this additional analysis to the manuscript since we wanted to avoid filling multiple pages of supplemental data with figures showing genome-wide analysis that failed to identify major differences between the wild type and *Setd2*-KO cells. We have now repeated the analysis including new single-cell RNA seq datasets, new analysis (alternative splicing and elongation ratios), and using the latest genome annotation, and have included all these results in the revised manuscript (New Figures 2, S7 and S8)

Major Point:

1) H3K36me3 and Alternative Splicing (Fig. 3C):

The authors report no correlation between differentially expressed genes and K36me3 deposition in *Setd2* KO day 8 NPCs. It would be valuable to analyze changes in alternative splicing (AS) of exons and introns where K36me3 levels are reduced. Single-end 150 bp RNA-seq should suffice to estimate AS changes.

This is particularly relevant because Neuronal tissues exhibit high AS diversity; SETD2 interacts with splicing factors like hnRNP L (PMID: 33664260); K36me3 regulates AS of splicing factors (e.g., SRSF11; PMID: 34480866); RNA Pol II elongation affects splicing decisions (PMID: 30988016).

Response #02: As suggested by the reviewer, we have now included the alternative splicing analysis in wild type and *Setd2*-KO neuronal progenitors using RNA seq with 125bp single end reads. We used rMATS-turbo (PMID: 38396040) for multivariate analysis of transcript splicing to identify statistically significant changes in alternative splicing events from mRNA molecules. The number of total and KO-/WT- specific splicing events are shown in table (Sup. Fig. S7). We observed 57 and 175 differential splicing events increasing and decreasing in *Setd2*-KO, respectively. However, these splicing differences are not correlated to changes in H3K36me3 levels around the spliced region (or entire gene body), suggesting that loss of H3K36me3 does not directly contribute to the observed effects (Sup Fig. S7D-E).

Gene expression analysis of splicing regulators shows that multiple factors involved in splicing regulation are differentially expressed in wild type and *Setd2*-KO neuronal progenitors (Sup. Fig. S7F), suggesting that the observed differential splicing could be an indirect consequence of global gene expression changes, including splicing regulators.

2) RNA Pol II ChIP-seq and Pausing Index:

TurboID-MS suggests reduced RNA Pol II elongation in SETD2 KO. The authors performed RNA Pol II ChIP-seq, which can serve to access the genome-wide pausing index at differentially expressed genes. If SETD2 affects Pol II elongation via PAF1C, changes in the pausing ratio (promoter vs. early

gene body) would be expected.

Response #03: We have now investigated RNAPII elongation rates in wild type and *Setd2*-KO cells in Fig. 4B and Sup. Fig S10. The RNAPII ChIP-seq data identified that in absence of *Setd2*, we observe reduced RNAPII levels at promoters and gene bodies when compared to wild type, especially at downregulated genes. RNAPII pausing index calculations did not reveal strong changes, however we observe slight reduction for downregulated genes.

3) *Ngn1/2* Overexpression and Transcriptional Initiation:

The authors speculate that *Ngn1/2* overexpression boosts PAF1C engagement with Pol II at neuronal genes, enforcing transcriptional initiation. However, *Ngn1/2* primarily acts on promoters and enhancers. The claim requires stronger evidence, as increased post-mitotic differentiation in NPC cultures could occur independently of PAF1C engagement.

Response #04: This is a misunderstanding that could be due to the wording we used in the initial version. We do not want to make the claim that *Ngn1/2* overexpression boosts “PAF1C engagement with Pol II”. What we indicate is that by increasing transcriptional initiation of neuronal genes (through *Ngn1/2* overexpression) we can help to partially overcome the transcriptional differences observed in absence of SETD2. Currently we state in the manuscript: “... *increasing the frequency of transcriptional initiation from neuronal genes through overexpressing neuronal master regulators can partially overrule the necessity for SETD2 in the differentiation process* “. We have rephrased this statement and have placed these experiments earlier in the manuscript to avoid further misunderstandings related to PAF1C (Now shown in Fig. 3).

4) Phenotypic Characterization of SETD2 KO Neurons (Fig. 1, Fig. 5):

Differentiation estimates are based on bright-field microscopy images, which lack quantitative detail. The authors should improve phenotypic characterization with immunofluorescence staining for neuronal markers. For defective cultures, how is neuronal survival estimated (Figs. 1B, C; 3B)? It's unclear if SETD2 KO leads to widespread cell death or a neurite growth defect. Quantifying these phenotypes during neuronal maturation, as previously reported (PMID: 28106138), would clarify the impact of SETD2 KO.

Response #05: In the previous version we counted the numbers of surviving neurons, starting from a specific number of neuronal progenitors plated on neuronal differentiation media. We have now improved the phenotypic characterization by additional approaches:

In a previous study, we have identified the two neuronal surface proteins, CD24a (CD24) and CD56 (also known as NCAM1) to be well-suited for quantitative assessment of neuronal lineage commitment of neuronal progenitors by FACS (PMID: 37385984). We have applied this approach to measure the neuronal differentiation in absence of SETD2 in ESCs and NPCs (new Sup. Fig S5H-I). While the amount of CD24+CD56 double-positive cells was similar in *Setd2*KO and WT cells (~ 24%), we observed a 14-15% reduction in CD24-positive cells in absence of SETD2 (50.6% in *Setd2*KO vs. 65.1% in WT) and increased numbers of double negative cells (24.2% vs. 11.4%, respectively). This quantification indicates that in absence of SETD2, ~15% of cells do not commit to neuronal differentiation and therefore do not survive the following steps of the neural differentiation protocol that promotes neuronal maturation.

To investigate if the surviving neurons in the *Setd2*-KO condition can fully mature, we have stained the mature neurons with Tuj1/MAP2, a well-established neuronal marker. We observe that the surviving *Setd2*-KO neurons are positive for Tuj1, indicating that they successfully produce mature neurons (New Sup. Fig. S3C).

In addition, analysis of bulk gene expression data indicates a potential delayed neuronal differentiation in absence of SETD2. Since the bulk analysis in the NPC state samples unsynchronized cells at different differentiation stages. we have now included single-cell RNA seq from wild type and *Setd2*-KO ES and NPC cells to obtain a more precise view on the gene expression changes during the differentiation process. From this new data, we observe 1) a clear differentiation trajectory of neuronal progenitor cells, and 2) that gene expression differences between *Setd2*-KO and wild type progenitor cells increases with progressing differentiation (new Figures 2C-E and Sup. Fig. S8).

5) The authors do not comment on the contribution to H3K36 methylation from NSD1-3 and ASHL1. It would be helpful to clarify in the introduction which methylation states are specifically deposited by SETD2.

Response #06: We have included this information in the introduction

Minor Points:

1) Why did the authors use the mm9 genome assembly instead of the more updated mm10 or mm39?

Response #07: At the time of the first submission, most of our in-house datasets and customary pipelines were available in mm9. We have now changed all analysis to mm39 – without resulting in any changes to the original results – and have updated all figures and analysis throughout the manuscript accordingly.

2) In Figure 4d, why weren't all proteins identified in Figure 4b included in the network reconstruction? Additionally, the meaning of "pooled background set" (line 189) is unclear.

Response #08: Figure 5D (previously 4d) shows only proteins that are differentially enriched between Setd2KO and WT, while Figure 5B (previously 4b) shows all proteins significantly detected by MS in the Setd2KO and in the WT sample over the pooled background sample, independent of their differential enrichment. We have now clarified this in the legend.

We have now better explained this "pooled background set" in the text. In brief, since the NLS-TurboID samples from wild type and Setd2 KO did not show any differences, we decided to combine all NLS-TurboID samples one background for the comparisons with SRI-TurboID in the wild type and Setd2-ko samples.

Reviewer #2 (Remarks to the Author):

In this manuscript, Ambrosi et al., investigate the role of histone and DNA methyltransferases in neuronal differentiation potential of mESCs, focusing on SETD2, a H3K36me3 methyltransferase. The authors demonstrate that SETD2 is important early on during the establishment of neuronal gene expression pattern but is dispensable once cells have differentiated. Further, they find that this is largely a non-catalytic function of SETD2, since the catalytic-dead enzyme was able to rescue differentiation. Mechanistically, using BioID experiments, they claim that SETD2 facilitates RNAPII interaction with the PAF1 complex, which is important for SETD2 mediated establishment of neuronal gene expression programs.

While the question being addressed and the initial observation that SETD2's catalytic activity is dispensable for differentiation are both interesting, the manuscript contains significant deficiencies that render it unsuitable for publication at this journal.

Major points:

1. The evidence for proposed mechanism is very rudimentary and technically problematic. First, using SRI domain fused to TurboID as a proxy to map RNAPII interactome, rather than tagging any RNAPII subunit, is a very surprising approach. Why didn't the authors simply tag one of the RNAPII subunits, like Rbp3? While it is known that SRI domain binds Ser2phos-Rbp1, one can't be sure of the localization dynamics of the overexpressed SRI-TurboID fusion. There is also evidence that the SRI can associate with DNA (PMID: 25925577).

Response #09: We think the reviewer did not fully consider the requirements for our experiment in their criticism. We wanted to specifically investigate the interactome of the elongating RNAPII on chromatin. Tagging RNAPII subunits directly with TurboID, as suggested by the reviewer, will not allow us to differentiate interactions between initiating, elongating, terminating, or even RNAPII subunits that are not associated with chromatin. This would drastically increase the background signal and would not help us identify interaction changes specific for the elongating Pol2. Therefore, we argue that the SRI-TurboID experiment is not a "proxy for mapping the RNAPII interactome", but that

the specificity of the SRI probe for Ser2-Phos-RNAPII allows us to exclusively probe protein interactions of the elongating RNAPII on chromatin.

To confirm the correct localization and activity of the SRI-TurboID probe, we performed co-immunostainings against Ser2-Phos-RNAPII and biotin in cell lines expressing SRI-TurboID or the unspecific NLS-TurboID probe. As can be seen from Sup. Fig. S11A-B, we observe a colocalization of biotin with the elongating RNA Pol2 only in the SRI-TurboID (colocalization coefficient = 0.7) but not in the NLS-TurboID cells (colocalization coefficient = 0.25). This is in line with previous probes we have developed in our group for similar applications (PMID: 32123383).

2. Second, the authors' claim that SETD2 promotes RNAPII interaction with PAF1 complex is not substantiated by the data. BioID mass-spec experiments are prone to a lot of variations and the differences in PAF1 complex members are very minimal and could be within the margin of error. Additionally, there is no validation of the BioID experiments using orthogonal approaches, such as co-IP. Without validation of BioID experiments, it is very hard to interpret any of the data.

Response #10: Please note that in our original BioID-MS experiments we included an unspecific NLS-TurboID probe to specifically account for variation and background, resulting in robust quantification of enrichments.

Following the request from the reviewer, we have tried to obtain the proteome of the elongating RNAPII on chromatin using an orthogonal approach. Towards this we performed a ChIP experiment where we first extracted formaldehyde-crosslinked chromatin from wild type and *Setd2*-KO neuronal progenitors and performed immunoprecipitation using antibodies specific for the elongating Ser2-Phos-RNAPII followed by mass spectrometry analysis of the IP material from 4 independent replicates. Unfortunately, the quality from this antibody-based ChIP-MS approach was comparable to results obtained from the TurboID, resulting in low recovery of identified proteins (Figure R1A). Nevertheless, the obtained results allowed us to explore the Ser2-Phos-RNAPII proteome in absence of SETD2. We again observe the PAF1c members *Ctr9*, *Cdc73* and *Leo1* only in the wild type interactome (Figure R1B and C), confirming our previous results obtained with SRI-TurboID.

Rebuttal Figure 1. ChIP-MS analysis in wild type and *Setd2*-KO neuronal progenitor cells using Ser2-Phos-RNAPII-specific antibodies. A) Number of Protein groups (after FDR) per raw file indicates low recovery of identified proteins. B) STRING analysis of RNAP2-S2P interactors enriched for wild type neuronal progenitors (over *Setd2*-KO NPC) identifies PAF1c members (*Ctr9*, *Cdc73*, *Leo1*). C) Bar plots representing the top ten Cellular Component GO terms summarizing the proteins enriched wild type and *Setd2*-KO NPCs respectively. Data was generated using the Enrichr database.

We agree that we do not have direct experimental evidence that SETD2 is physically promoting RNAPII interactions with PAF1c, and that the differences in RNAPII interactions with PAF1c in absence of SETD2 could be completely indirect. However, we provide multiple lines of evidence that in absence of SETD2 the interactions between RNAPII and PAF1 are weakened. In addition, while preparing the revisions for this manuscript, a preprint was posted where the authors performed a MS-based proteome analysis of nuclear extracts from wild type and *Setd2* KO U2OS cells (doi: <https://doi.org/10.1101/2025.07.14.664659>). The authors observed several proteins associated with all steps of RNAPII activity, including PAF1c members, to be depleted in the *Setd2*-KO cells, supporting the notion that SETD2 could have indirect effects on RNAPII activity. We have now

changed the text in the manuscript to avoid any misinterpretation and state clearly that the PAF1C changes that we observe in absence of SETD2 could be indirect.

3. Third, the authors' claim that the observed phenotype in *Setd2*-KO cells is due to reduced PAF1 complex with RNAPII is also not supported by their data. The attempt to rescue *Setd2*-KO cells by overexpressing neuronal transcription factors doesn't prove that SETD2 acts through promoting PAF1 association with RNAPII. Overexpression of such master TFs could potentially override any number of molecular manipulations and it is almost impossible to categorically point to one specific mechanism.

Response #11: We agree that the *Ngn1/2* overexpression experiments do not provide evidence for SETD2's role in promoting PAF1C interactions with RNA pol 2 (see also Response #4). We have rephrased this in the text to highlight that increasing transcriptional initiation of neuronal genes (through *Ngn1/2* overexpression) can help to partially overcome the transcriptional differences observed in absence of SETD2 – independent of any potential interactions with PAF1c. Furthermore, we placed these experiments earlier in the manuscript to avoid further misunderstandings (Now shown Fig. 3).

4. The molecular characterization of differentiation is weak. The only read out the authors use for differentiation is the percentage of cells that survive the differentiation protocol. However, there is no attempt to properly characterize whether the surviving cells exhibit properties of properly differentiated neuronal cells, except for a cursory attempt in the last figure.

Response #12: As also replied in response #5, we have now included additional measurements of neuronal differentiation:

- 1) We measured the presence of two neuronal surface proteins, CD24a (CD24) and CD56 (also known as NCAM1) (PMID: 37385984), by flow cytometry as a quantitative assessment of neuronal lineage commitment of neuronal progenitors. This identified an increase in double negative cells in absence of SETD2 indicating lack of commitment to neuronal differentiation (new Sup. Fig. S5H-I).
- 2) To investigate if the surviving neurons in the *Setd2*-KO condition can fully mature, we have stained the mature neurons with Tuj1/MAP2, a well-established neuronal marker. We observe that the surviving *Setd2*-KO neurons are positive for Tuj1, indicating that they successfully produce mature neurons (new Sup. Fig. S3C).
- 3) We have now included single-cell RNA seq from wild type and *Setd2*-KO ES and NPC cells to obtain a more precise view on the gene expression changes during the neuronal differentiation process. From this new data, we observe 1) a clear differentiation trajectory of neuronal progenitor cells, and 2) that gene expression differences between *Setd2*-KO and wild type progenitor cells increases with progressing differentiation (new Fig 2C-E and Sup. Fig S8).

5. While the CRISPR mediated R1599H point mutant is reasonable, the authors should have also used the SETD2 inhibitor (EPZ-719) to orthogonally test if SETD2's catalytic activity is required for differentiation.

Response #13: We have tested the SETD2 inhibitor EPZ-719 in the differentiation process to evaluate if we can observe similar phenotypes as observed with the full KO. For this we grew wild type ES cells in DMSO and 500nM EPZ-719 for two days before we started the neuronal differentiation, maintaining the drug concentrations throughout the entire differentiation procedure until the terminal neuron stage and recording of survival. We could not observe any effects of the drug on the differentiating neurons, supporting that the catalytic activity is not required (New Sup. Fig. S10J).

This is also in line with a recent preprint that compares the effect of full *Setd2*-KO to EPZ-719, where they observe that despite H3K36me3 reduction in the drug treatment, the resulting changes in transcription readthrough is not comparable with the full KO – indicating H3K36me3-independent roles of SETD2 (doi: <https://doi.org/10.1101/2025.07.14.664659>).

6. The ChIPseq data in Fig 3a are very noisy and also not well analyzed. There's no global analysis of all the data except for H3K36me3 and H4-panAc

Response #14: We have included additional global data analysis in Sup. Fig. S10

Overall, the manuscript in its current form is in a very elementary stage and many of the author's claims are not substantiated by the data and therefore it would take an enormous amount of work to meet the standards of NSMB for publication.

Reviewer #3 (Remarks to the Author):

In the manuscript "SETD2 promotes PAF1C interactions with the elongating RNA Pol II and is required for neuronal differentiation" Ambrosi et al. set out to compare the effects of depleting key chromatin factors by CRISPR/Cas9 in isogenic mES cell lines on neuronal differentiation. I consider this unification effort important, as often in the chromatin field work is focused on one factor and compared to results obtained using a model generated in a different lab using another background. The neuronal differentiation experiments showed, that the various chromatin regulators tested (EED, DNMT1, DNMT3A, DNMT3B, and SETD2) were required at different stages of cellular differentiation. While EED and DNMTs were required early in the differentiation protocol (Eed-KO cells did not survive exit from pluripotency and Dnmt-TKO cells failed differentiation at the neural progenitor stage), SetD2, the enzyme responsible for H3K36me3, was important during late stages of differentiation, for the establishment of proper neuronal gene expression. Importantly, the authors reproduced their results in two independent knock-out clones and one constitutive Setd2 knock-down cell line. This is in line with mouse KO model which previously showed that Setd2^{-/-} is embryonically lethal only at day E10.5, leading to growth defects including forebrain hypoplasia and unclosed neural tubes (Hu, M. et al. PNAS 2010). Further on, the authors demonstrated in the current manuscript using SETD2 depletion in a dox-inducible system that SETD2 was dispensable once the cells have fully differentiated. The authors then checked whether the lysine methyltransferase activity of SETD2 is important for this function, by generating a methyltransferase catalytic-dead mutant (R1599H) knock-in. Surprisingly, this had only a mild effect, implying that the function of SETD2 in neuronal differentiation is mostly independent of its catalytic activity. The authors employed then an elegant approach, they established in their own lab previously, ChromID, to measure the protein interaction network of elongating RNA Pol II. They fused the SRI domain of SETD2, which binds the serine-2-phosphorylated CTD of RNA Pol II, to the promiscuous biotin ligase TurboID. This enabled them identification of proteins associated with the elongating RNA Pol II by proximity biotinylation. This revealed that multiple subunit of the transcription elongation complex PAF1 were less associated to elongating RNA Pol II. The authors concluded that they identified a novel role for SETD2 in mediating interactions between the PAF1 complex and the elongating RNA Pol II, which is required to ensure optimal transcriptional processivity of neuronal genes, and independent of its catalytic activity.

I find the presented approaches valid and quality of data high. The conclusions are well justified by the data. Besides the important insights obtained directly from the study, the obtained isogenic cell lines will be a valuable resource for the chromatin community

My feeling is that the results presented are of immediate interest to many people in the transcription and chromatin fields, who are a significant fraction of readers of NSMB, and potentially also to biomedical researchers working on epigenetic therapies.

I think the length of the text is appropriate to describe the results. I would only suggest to expand the discussion (slightly) and methods (more) sections, see points below.

Overall, I recommend this manuscript for publication in NSMB.

Nevertheless, there are several small but important issues that need to be revised in my view.

1. Most importantly – ChromID methodological description.

I find the modular building blocks approach of the ChromID method very elegant. In particular, analysing RNA Pol II interactome by proximity biotinylation has many advantages over more traditional immunoprecipitation-based methods. However, as this is still a novel method, and not standard in the field, it should be much better described methodologically, in particular in respect to the analysis. Ideally a script should be made available to allow full reproducibility of the results obtained. Minimally, much more details of the approach, parameters used etc.

Response #15: We thank the reviewer for this constructive comment. The method itself is described in Villasenor et al Nat Biotech 2020 (PMID: 32123383). We have now added the scripts used to analyze the MS data to github and refer to it in the methods and data availability section.

2. Similarly, a script for the NGS data analysis in github or zenodo would be helpful for reproducibility.

Response #16: We have now included these scripts to github, mentioned in the methods and data availability section.

3. Since the authors performed RNA-seq in wt and SETD2 KO conditions, did they see any indication of missplicing? This appears to occur in some models in literature but not all.

Response #17: We have now analyzed splicing differences between Setd2-KO and wt cells and do not see drastic differences that are associated with loss of H3K36me3. Please see response #2 and new Sup. Fig. S7.

4. I would like to see more information/discussion about non-PAF1 factors found in ChromID. Eg, KMT2E appears the most depleted (fig S8B) in SETD2 KO, could the authors comment on this? Figure 4B – does the high SETD2 values in the SETD2 KO come from the SRI autobiotinylation? Please indicate this in manuscript.

Response #18: We mention non-Paf1 complex members in the text now. Indeed, the high SETD2 values come from the autobiotinylated SRI peptides. We have included this information in the legend.

5. Figure S6f – in the authors' opinion, does the remaining H3K36me3 signal in SETD2 KO correspond more likely to residual H3K36me3 or cross-reactivity of the antibody with K36me2? I would suggest to openly mention this in the manuscript.

Response #19: Yes, the antibody used has a residual affinity for H3K36me2. We mention this now in the figure legend of now Fig. S10I.

6. Fig 3b Authors say “modest correlation”, but indicate no p-values. I assume the number on the graph is a correlation coefficient, but this is not explained in the legend.

Response #20: We have added this missing information to the legend.

Reviewer #4 (Remarks to the Author):

Ambrosi et al explore the role of the H3K36 trimethyltransferase Setd2 during exit from pluripotency and differentiation into neural progenitor cells (NPCs) and neurons using mouse ESCs. They find that while Setd2 is dispensable for pluripotency exit and NPC generation, it is required for the establishment but not maintenance of neurons. They further show that this phenotype is driven by a non-catalytic function of Setd2. Proximity labeling combined with mass spec suggests that Setd2's role in differentiation may be mediated by the association of Pol2Ser2 with the Paf1 complex to safeguard transcriptional processivity of neuronal targets. Consistent with this idea, overexpression of a neuronal TF partially rescues the differentiation defect of Setd2^{-/-} cells.

Overall, this is a well-performed study that provides insights into the non-catalytic role of Setd2 in neuronal differentiation, which appears to be mediated at least in part via the Paf1 complex implicated in transcriptional elongation. However, the authors do not directly show that there is a defect in transcriptional processivity. Moreover, several of the presented results are confirmatory of previous observations, such as the differentiation defects of Eed, Dnmt TKO cells and the lack of a strong correlation between loss of H3K36me3 in Setd2 KO cells and gene expression changes. The manuscript in its current form may therefore be a better fit for a more specialized journal.

Specific points:

1. Survival is not a specific assay to infer neuronal differentiation defects. The authors need to

perform IF for appropriate markers in control and all mutant cells to conclusively show that there is indeed a differentiation defect.

Response #21: As also replied in response #5, we have now included additional measurements of neuronal differentiation, including markers:

- 1) We measured the presence of two neuronal surface proteins, CD24a (CD24) and CD56 (also known as NCAM1) (PMID: 37385984), by flow cytometry as a quantitative assessment of neuronal lineage commitment of neuronal progenitors. This identified an increase in double negative cells in absence of SETD2 indicating lack of commitment to neuronal differentiation (new Sup. Fig. S5H-I).
- 2) To investigate if the surviving neurons in the *Setd2*-KO condition can fully mature, we have stained the mature neurons with Tuj1/MAP2, a well-established neuronal marker. We observe that the surviving *Setd2*-KO neurons are positive for Tuj1, indicating that they successfully produce mature neurons (new Sup. Fig. S3C).
- 3) We have now included single-cell RNA seq from wild type and *Setd2*-KO ES and NPC cells to obtain a more precise view on the gene expression changes during the neuronal differentiation process. From this new data, we observe 1) a clear differentiation trajectory of neuronal progenitor cells, and 2) that gene expression differences between *Setd2*-KO and wild type progenitor cells increases with progressing differentiation (new Fig. 2 and Sup. Fig. S8).

2. The authors' conclusion that *Setd2* is not required for neuronal maintenance is interesting. However, they need to confirm that *Setd2* is still efficiently depleted by Western blot analysis (i.e. is the hairpin vector still expressed?) at the end point of differentiation to rule out silencing of the vector.

Response #22: In Sup Fig S5A (previous S4c) we show depleted *Setd2* mRNA levels in presence of shRNA expression in terminal neurons. Furthermore, we have now repeated these experiments with a new cell line where we have tagged both endogenous alleles of SETD2 with FKBP, allowing controlled degradation the protein via addition of dTAG13. These additional experiments show identical differentiation results to those obtained with the shRNA approach (Sup Fig. S5).

3. The authors should perform RNA-Seq of the catalytically dead *Setd2* cells before and after differentiation and compare with WT and *Setd2* KO cells to better appreciate how the catalytic and non-catalytic roles of *Setd2* affect transcriptional output globally.

Response #23: We have first tested the catalytic activity of SETD2 during neuronal differentiation experiments using an orthogonal approach - as suggested by Reviewer #2. Here we made use of the novel SETD2 inhibitor EPZ-719 (PMID 34671445) and performed the differentiation in presence of this inhibitor. Similar to the results obtained from the catalytic mutant, we did not observe severe changes in neuronal progression, supporting that catalytic activity is dispensable for neuronal differentiation.

Following these results, and since we have not observed any association of H3K36me3 with changes in gene expression or alternative splicing in the full SETD2 KO NPC cells (Sup Fig. S7 and S10), we argued that including RNA-seq analysis of the catalytic mutants in ESCs and NPCs would not add additional information during the revision process, where we focused our attention on validating the initial results with new cell lines (*Setd2*-dTAG) and at higher resolution (single-cell RNA seq). We hope the reviewer agrees that prioritizing these experiments over the RNA-seq analysis in the catalytic mutant will add more information to the final manuscript.

4. The authors should consider doing Gro-Seq or Pro-Seq in WT and mutant cells to directly test whether there are defects in transcriptional processivity, as proposed by their model.

Response #24: We have analyzed the RNA Pol 2 Chip-seq data to test potential defects on transcriptional processivity. Here, as expected, we observed decreased initiation and elongation signals at genes down-regulated in absence of *Setd2*, while we did not see drastic changes in RNA P2 pausing rates (Figure 4B and Sup. Fig. S10F-G). In addition, also based on comments from Reviewer #2, we have now changed the main text to carefully indicate that we do not have direct evidence on changes in transcriptional processivity in absence of SETD2, and furthermore to mention

that the reduced interactions of PAF1c and RNA Pol 2 could be of indirect nature. These would need to be followed up with more rigorous molecular and biochemical approaches in future studies, which are beyond this current manuscript. We hope the reviewer agrees to these changes.

We would also like to point to a recent publication that has performed nascent transcriptomics in Setd2-KO cells (<https://doi.org/10.1101/2025.07.14.664659>), revealing gene-to-gene-specific changes in transcriptional profiles, with either reduced transcriptional initiation or transcriptional readthrough. This highlights the complex transcriptional responses in absence of SETD2, which will require extensive effort to fully disentangle mechanistically. We hope to address these in the next study.

5. The authors need to include a WT minus Dox control in their rescue experiment in Fig. 5c to assess whether candidate gene expression, particularly Ngn1/2, is different in Setd2 KO cells. If Ngn1/2 levels are down in Setd2 KO cells, this could be a more trivial explanation to explain the differentiation defect and the authors' ability to rescue the phenotype via Ngn1/2 overexpression. Do the authors expect that other neuronal TFs have a similar effect, i.e. is there some specificity to Ngn1/2?

Response #25: We have investigated whether Ngn1/2 are differentially expressed in Setd2 vs. Wild type cells. While we identify several neuronal genes downregulated in the KO, Ngn1/2 are not among them. Furthermore, we have now performed single-cell RNA seq to obtain a better view on the differentiation trajectories and their differences between wild type and Setd2 KO NPCs. Same as with the bulk RNA-seq data, while we find many neuronal transcription factors and regulators to be differentially expressed (Fig 2C-E and Sup Fig S8), we do not identify Ngn1/2 among those with statistically significant changes.

The reason we used Ngn1/2 is predominantly of technical nature. These factors have been previously shown to promote faster neuronal differentiation (PMID:25403753) and furthermore used to force neuronal differentiation in other epigenetic mutants (DNA methylation) using the same differentiation protocol (PMID: 36471082).

6. Statistical analyses are missing in several places, such as Figs. 1b, 3d, 5c, etc. and need to be provided.

Response #26: We have included the statistical tests as requested.

7. Zhang et al previously showed that Setd2 is required for ESC differentiation towards primitive endoderm (PMID: 25242323). The authors should discuss this finding in the context of their results.

Response #27: - This information is now included in the manuscript

8. Do the different colors in Fig. 4d have any particular meaning?

Response #28: No, we have now removed them from the figure to avoid confusion.

Dear Tuncay

Thank you again for submitting your manuscript for consideration at The EMBO Journal. Please find enclosed the comments from two of the reviewers who evaluated the revised version. The remaining two original referees were ultimately not able to re-review the manuscript. As you will see from the reports, both referees are now generally supportive; however, referee #2 still feels that the mechanistic insight is limited and that the manuscript would benefit from additional analysis of the existing datasets. I consider this a productive suggestion and find the report overall constructive. I would therefore like to invite you to revise the manuscript in line with these recommendations. Please do not hesitate to contact me if you have any further questions.

With best regards,

Cornelius

Yours sincerely,

Cornelius Schneider, PhD
Editor
The EMBO Journal
c.schneider@embojournal.org

Read our guidance for manuscript revisions and related editorial policies: <https://link.springer.com/journal/44318/submission-guidelines#cms-Revised-submissions>

<https://media.springernature.com/original/springer-cms/rest/v1/content/27825798/data/v1>

- a point-by-point response to the referees' comments, with a detailed description of the changes made (as a word file).
- a word file of the manuscript text.
- individual production quality figure files (one file per figure)
- a complete author checklist
- Expanded View files (replacing Supplementary Information)
- a Reagents and Tools Table as part of the Methods section

Please remember: Digital image enhancement is acceptable practice, as long as it accurately represents the original data and conforms to community standards. If a figure has been subjected to significant electronic manipulation, this must be noted in the figure legend or in the 'Methods' section. The editors reserve the right to request original versions of figures and the original images that were used to assemble the figure.

We realize that it is difficult to revise to a specific deadline. In the interest of protecting the conceptual advance provided by the work, we recommend a revision within 3 months (11th Mar 2026). Please discuss the revision progress ahead of this time with the editor if you require more time to complete the revisions. Use the link below to submit your revision:

Referee #1:

In the revised manuscript "SETD2 contributes to PAF1C interactions with the elongating RNA Pol II and is required for neuronal differentiation" the authors have fully satisfactorily responded to the points I raised in the first review (response #15-20). In

particular, I confirm the scripts used for the analysis have been deposited in github, and are well described with ample comments.

I find the new alternative splicing analysis performed and reported (in response to reviewer 1 and myself) of good quality and very important for a wide audience (response #2 and #17). Reassuringly, our laboratory has also found no evidence for direct regulation of alternative splicing events by this factor, using two independent SETD2 depletion models (unpublished). It is important to correct this misconception in the field.

Besides the revised manuscript, I've read the responses to all the other concerns raised by the other 3 reviewers and find the authors' answers well-founded and convincing. I appreciate the more careful characterization of the neuronal differentiation performed, new single-cell RNA-seq data, the use of an fully orthogonal SETD2 degron cell line, as well as the specific inhibitor EPZ-719, all supporting and strengthening the original conclusions of the manuscript.

I stand by my previous assessment that the results presented are of immediate interest to many people in the transcription and chromatin fields. In my opinion this revised work is suitable for publication in EMBO Journal.

Referee #2:

Ambrosi et al. investigate the role of SETD2 in neurodevelopment, proposing that it functions independently of its canonical enzymatic activity-H3K36me3 deposition. This conclusion is supported by the observed lack of correlation between gene expression and H3K36me3 levels, as well as by the ability of catalytically inactive SETD2 mutant lines to differentiate into mature neurons. The authors further strengthen their argument with degron-mediated SETD2 depletion and inhibitor-based experiments, both of which robustly demonstrate that SETD2 is required for neuronal differentiation. These experiments are well-designed and convincingly establish the relevance of SETD2 in neurodevelopment.

As a potential catalytic activity-independent mechanism, the authors employ TURBO-ID proximity labeling and identify multiple SETD2-interacting proteins. They focus particularly on the PAF1 complex and propose that the SETD2-PAF1 interaction contributes to the establishment of neuronal gene expression programs. This is an interesting direction that expands the mechanistic landscape of SETD2 function beyond methyltransferase activity.

While the overall findings are significant, the study's mechanistic depth remains limited. The data interpretation, especially in the ChIP-seq and ATAC-seq analyses, appears overly simplified. For example, H3K27me3 levels increase in SETD2 knockout cells (Appendix Figure S10A), but it is unclear whether this indicates a global shift or gene-specific enrichment. Since H3K27me3 is a key repressive mark regulating neurodevelopmental gene expression, such alterations could contribute to the observed differentiation defects, either independently or synergistically. The authors should explore or at least discuss this possibility, as it may substantially affect the interpretation of SETD2's role in chromatin regulation.

Also, it would be beneficial to include a schematic model summarizing the proposed catalytic-independent mechanisms of SETD2. Such a figure could illustrate how SETD2 integrates with the PAF1 complex and chromatin states (e.g., H3K36me3 and H3K27me3) to coordinate transcriptional programs during neurodevelopment.

In summary, this study provides an important refinement of SETD2's role in neuronal differentiation and raises compelling questions about its non-catalytic functions. However, the mechanistic framework is not yet fully convincing. Stronger integration of chromatin data and a more comprehensive analysis of alternative epigenetic effects (e.g., H3K27me3 changes, if any) would enhance the impact and interpretability of the work.

Point by point response to reviewer's comments EMBOJ-2025-120622R**Referee #1:**

In the revised manuscript "SETD2 contributes to PAF1C interactions with the elongating RNA Pol II and is required for neuronal differentiation" the authors have fully satisfactorily responded to the points I raised in the first review (response #15-20). In particular, I confirm the scripts used for the analysis have been deposited in github, and are well described with ample comments.

I find the new alternative splicing analysis performed and reported (in response to reviewer 1 and myself) of good quality and very important for a wide audience (response #2 and #17). Reassuringly, our laboratory has also found no evidence for direct regulation of alternative splicing events by this factor, using two independent SETD2 depletion models (unpublished). It is important to correct this misconception in the field.

Besides the revised manuscript, I've read the responses to all the other concerns raised by the other 3 reviewers and find the authors' answers well-founded and convincing. I appreciate the more careful characterization of the neuronal differentiation performed, new single-cell RNA-seq data, the use of an fully orthogonal SETD2 degron cell line, as well as the specific inhibitor EPZ-719, all supporting and strengthening the original conclusions of the manuscript.

I stand by my previous assessment that the results presented are of immediate interest to many people in the transcription and chromatin fields. In my opinion this revised work is suitable for publication in EMBO Journal.

Response #1 – We thank the reviewer for the positive comments and support.

Referee #2:

Ambrosi et al. investigate the role of SETD2 in neurodevelopment, proposing that it functions independently of its canonical enzymatic activity-H3K36me3 deposition. This conclusion is supported by the observed lack of correlation between gene expression and H3K36me3 levels, as well as by the ability of catalytically inactive SETD2 mutant lines to differentiate into mature neurons. The authors further strengthen their argument with degron-mediated SETD2 depletion and inhibitor-based experiments, both of which robustly demonstrate that SETD2 is required for neuronal differentiation. These experiments are well-designed and convincingly establish the relevance of SETD2 in neurodevelopment.

As a potential catalytic activity-independent mechanism, the authors employ TURBO-ID proximity labeling and identify multiple SETD2-interacting proteins. They focus particularly on the PAF1 complex and propose that the SETD2-PAF1 interaction contributes to the establishment of neuronal gene expression programs. This is an interesting direction that expands the mechanistic landscape of SETD2 function beyond methyltransferase activity.

While the overall findings are significant, the study's mechanistic depth remains limited. The data interpretation, especially in the ChIP-seq and ATAC-seq analyses, appears overly simplified. For example, H3K27me3 levels increase in SETD2 knockout cells (Appendix Figure S10A), but it is unclear whether this indicates a global shift or gene-specific enrichment.

Since H3K27me3 is a key repressive mark regulating neurodevelopmental gene expression, such alterations could contribute to the observed differentiation defects, either independently or synergistically. The authors should explore or at least discuss this possibility, as it may substantially affect the interpretation of SETD2's role in chromatin regulation.

Response #2 – We would like to point out that the difference in H3K27me3 levels in Appendix Figure S10A stem from the different scaling of the wild type and Setd2-KO tracks (due to autoscaling). Our analysis of H3K27me3 did not reveal significant differences at gene promoters or gene bodies of genes that change gene expression in absence of Setd2, and at gene targets that lose H3K36me3 in the same cell line. This strongly suggests that H3K27me3 does not play a role in the described phenotypes. Some of these results were already present in the previous manuscript (Appendix Figures S10C and D). To follow the reviewer's suggestions, we have repeated our analysis of genome-wide H3K27me3 datasets to explore the possibility that H3K27me3 changes could be

related to differential gene expression or loss of H3K36me3 in NPCs. In this new figure (Appendix Figures S10E), where we also expand the analysis to additional histone modifications and ATAC seq, we again observe that, except for a slight correlation between accessibility and histone acetylation, changes in transcription are not correlated to changes in H3K27me3 at gene promoters. Furthermore, the genes that show changes in gene expression, have very similar H3K27me3 distribution over their gene promoters (Appendix Figures S10F).

Similar to the H3K27me3 example, we have performed extensive analysis of RNA-, ATAC, and ChIP-seq datasets generated throughout this study to identify potential correlations with H3K36me3 loss or transcriptional changes. Besides the observed correlated and expected changes of ATAC seq signals, Histone acetylation, and RNA Pol 2 occupancy at the promoters of differentially expressed genes (already present in Fig 4B-D and in Appendix Figures S10C-D), we failed to identify major chromatin differences between the wild type and *Setd2*-KO cells in all datasets. This indicates that loss of H3K36me3 results in limited to no chromatin changes, which is also visible by the good genome-wide pairwise correlations between WT and KO samples of the same dataset, with H3K36me3 being the only exception as expected (shown in Appendix Figure S10B). We mention this in the manuscript, and we prefer to avoid filling the manuscript with plots that don't add additional information. We hope the reviewer agrees with this.

Also, it would be beneficial to include a schematic model summarizing the proposed catalytic-independent mechanisms of SETD2. Such a figure could illustrate how SETD2 integrates with the PAF1 complex and chromatin states (e.g., H3K36me3 and H3K27me3) to coordinate transcriptional programs during neurodevelopment.

Response #3 – We have added a schematic model to appendix figure S13E summarizing the conclusions drawn from this study.

In summary, this study provides an important refinement of SETD2's role in neuronal differentiation and raises compelling questions about its non-catalytic functions. However, the mechanistic framework is not yet fully convincing. Stronger integration of chromatin data and a more comprehensive analysis of alternative epigenetic effects (e.g., H3K27me3 changes, if any) would enhance the impact and interpretability of the work.

Dear Prof. Baubec,

Thank you for submitting a revised version of your manuscript. Your study has now been seen by the referee #2, who finds that their previous concerns have been addressed and now recommends publication of the manuscript. There remain only a few mainly editorial points that have to be addressed before I can extend formal acceptance of the manuscript:

- As we are switching from a free-text author contribution statement towards a more formal statement based on Contributor Role Taxonomy (CRediT) terms, please remove the present Author Contribution section and instead specify each author's contribution(s) directly in the Author Information page of our submission system during upload of the final manuscript. See <https://casrai.org/credit/> for more information.

- Please rename the Conflict of Interest section into "Disclosure and Competing Interests Statement", in accordance with our updated Guide to Authors (<https://link.springer.com/partners/embo-press/editorial-policies#Competing%20interest%20disclosures>)

- FIGURE CALLOUTS: missing for Figure 5B; Appendix Fig. 9B should be corrected to Appendix Fig. S9B; Appendix (Supplementary) table callouts should be renamed to Dataset EV1-EV4

- DATASET EV LEGENDS: source file names, titles, legends and manuscript callouts all need to be updated to Dataset EV1-EV# instead of Appendix Table S1-S4, legends should be removed from Appendix PDF and uploaded as a separate tab/sheet in each Excel file

- APPENDIX 1 FILE WITH ToC: title page should contain "Appendix for + manuscript title" and ToC with the page numbers for the listed items; Appendix Table legends should be removed from Appendix PDF

- SOURCE DATA: Source data files need to be saved in a scheme one figure/folder and then uploaded as .zip files. E.g. all the Source data files for figure 1 need to be saved in a single folder and this needs to be zipped and then uploaded as "SD figure 1.zip" file. For EV and/or appendix figures, ZIP together all source data. Completed SD checklist should be uploaded separately as Related Manuscript File.

- Please provide suggestions for a short 'blurb' text prefacing and summing up the conceptual aspect of the study in two sentences (max. 250 characters), followed by 3-5 one-sentence 'bullet points' with brief factual statements of key results of the paper; they will form the basis of an editor-written 'Synopsis' accompanying the online version of the article. Please also provide an altered synopsis image, making sure that the aspect ratio conforms to our website's format - it should be exactly 550 pixels wide and between 300-600 pixels high.

- Figure Legends - Comments:

- Please define the annotated p values ****/***/**/* as well as provide the exact p-values for the same in the legend of figure 4E as appropriate.

- Please note that the exact p values are not provided in the legend of figure 1A

- Please indicate the statistical test used for data analysis in the legends of figures 4E

- Please note that scale bar and its definition are missing for figures 1B, C

- Sections need to be named and the order should be corrected: Title page - Abstract - Introduction - Results - Discussion - Methods - Data Availability - Acknowledgements - Disclosure and Competing Interests Statement - References - Figure Legends - Table(s) - Expanded View Figure Legends.

With best regards,

Cornelius Schneider

Cornelius Schneider, PhD
Editor | The EMBO Journal
c.schneider@embojournal.org

Please refer to our figure preparation guideline in order to ensure proper formatting and readability in print as well as on screen:

<https://link.springer.com/journal/44318/submission-guidelines#cms-Figure-and-data-presentation>

Use the link below to submit your revision:

Referee #2:

I appreciate the authors' efforts in thoroughly addressing my previous comments. They have provided clear responses to all points raised. In my opinion, the revisions have strengthened the manuscript, and the work is now suitable for publication in The EMBO Journal.

All editorial and formatting issues were resolved by the authors.

Dear Prof. Baubec,

I am pleased to inform you that your manuscript has been accepted for publication in the EMBO Journal.

You may qualify for financial assistance for your publication charges - either via a Springer Nature fully open access agreement or an EMBO initiative. Check your eligibility: <https://link.springer.com/journal/44318/how-to-publish-with-us>

Yours sincerely,

Cornelius Schneider, PhD
Editor
The EMBO Journal
c.schneider@embojournal.org

Please note that it is The EMBO Journal policy for the transcript of the editorial process (containing referee reports and your response letters) to be published as an online supplement to each paper. If you should prefer removal of any referee-only figures included in the point-by-point response(s), e.g. because they may still be used for future publication or because they have been reproduced from published work by others, please do let us know immediately via response email.

More information is available here: <https://link.springer.com/partners/embo-press/editorial-policies#Peer%20review>
